Subject Areas:
microbiology/molecular biology/biochemistry

Keywords:
*Trypanosoma*, *Leishmania*, mRNA decay, mRNA translation, mRNA processing, transcription

Author for correspondence:
Christine Clayton
e-mail: cclayton@zmbh.uni-heidelberg.de

# Regulation of gene expression in trypanosomatids: living with polycistronic transcription

## Christine Clayton

University of Heidelberg Center for Molecular Biology (ZMBH), Im Neuenheimer Feld 282, D69120 Heidelberg, Germany

 CC, 0000-0002-6384-0731

In trypanosomes, RNA polymerase II transcription is polycistronic and individual mRNAs are excised by *trans*-splicing and polyadenylation. The lack of individual gene transcription control is compensated by control of mRNA processing, translation and degradation. Although the basic mechanisms of mRNA decay and translation are evolutionarily conserved, there are also unique aspects, such as the existence of six cap-binding translation initiation factor homologues, a novel decapping enzyme and an mRNA stabilizing complex that is recruited by RNA-binding proteins. High-throughput analyses have identified nearly a hundred regulatory mRNA-binding proteins, making trypanosomes valuable as a model system to investigate post-transcriptional regulation.

## 1. Introduction

In trypanosomatids, initiation of transcription by RNA polymerase II is not controlled at the level of individual genes. Instead, there is regulation of mRNA processing, translation and decay. Trypanosomatids that are easily cultured and genetically manipulated are, therefore, excellent models for the study of post-transcriptional control. The supergroup to which trypanosomatids belong, Excavata, diverged very early in eukaryotic evolution from the supergroups that include mammals, fungi and plants [1] (electronic supplementary material, Note 1 in Supplementary Notes). Because of this, comparisons between the different models enable us to distinguish characteristics that were present in the last eukaryotic common ancestor from those that evolved later [2]. Moreover, one aspect of trypanosome gene expression—their mRNA processing—is a target of clinically useful drugs for the treatment of human and ruminant trypanosomiasis [3,4].

In this review, I summarize our knowledge of kinetoplastid RNA polymerase II transcription, mRNA processing and export, then I describe regulation of mRNA translation and mRNA degradation in more detail. The electronic supplementary material, Notes, provides extra information and references on particular topics, while electronic supplementary material, table S1, lists many of the proteins involved. The table includes details of all currently known mRNA-binding proteins, and other proteins that have been implicated in post-transcriptional RNA control.

Trypanosomatida are intracellular or extracellular parasites of plants and/or animals [5]. All have a single flagellum and a complex of circular mitochondrial DNAs called the kinetoplast. Trypanosomatid parasites of medical, veterinary or agricultural importance are usually transmitted by arthropods; they include plant parasites of the genus *Phytomonas*, and *Leishmania* and *Trypanosoma* species, which cause diseases in vertebrates.

Gene expression has been studied most extensively in the African trypanosome *Trypanosoma brucei*, which will, therefore, be the main focus of this review. *T. brucei* live extracellularly in the blood and tissue fluids of mammals, and in the digestive system of tsetse flies. Within mammals, the long slender

royalsocietypublishing.org/journal/rsob   Open Biol. 9: 190072

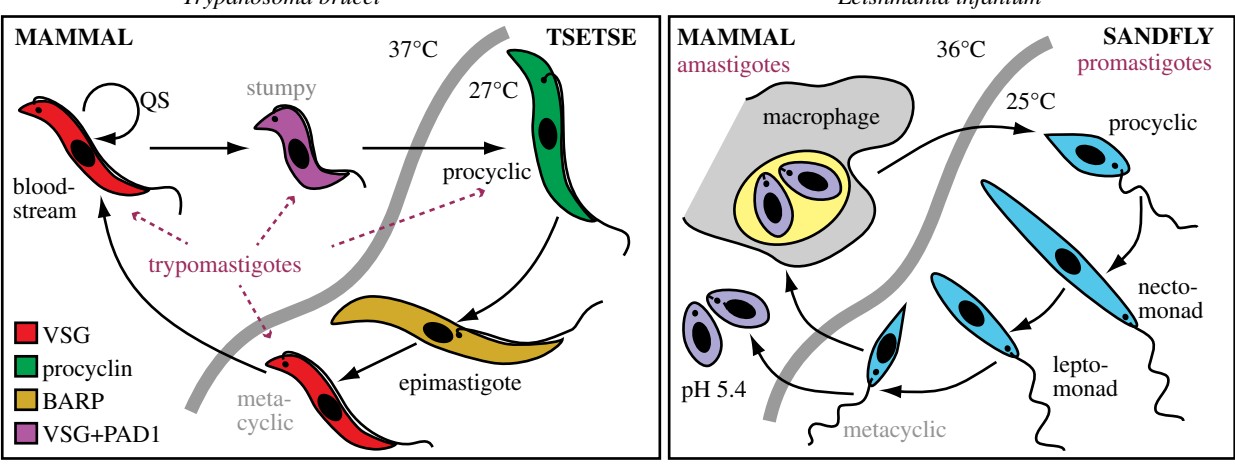

**Figure 1.** Life cycles of *T. brucei* and *Leishmania*. These simplified diagrams show the main developmental stages mentioned in this review. Different surface molecules are colour coded. The different stages are named according to the presence or absence of an external flagellum, overall shape and the position of the kinetoplast relative to the nucleus. In trypomastigotes (dotted arrows), the kinetoplast and flagellar base are posterior to the nucleus; in promastigotes, they are anterior. Amastigotes are amotile and have only a flagellar stub. Non-dividing transmission forms are labelled in grey and the macrophage lumen is pale yellow; QS means quorum sensing. Transformation of bloodstream-form trypanosomes to procyclic forms can be induced *in vitro* by applying specific stimuli, such as *cis*-aconitate, and changing the culture conditions (providing proline, decreasing the temperature). (*b*) Transformation of *Leishmania* metacyclic promastigotes to axenic amastigote-like forms can be achieved by elevating the temperature and the medium acidity. Culture temperatures for amastigotes are species dependent.

bloodstream-form trypomastigotes evade humoral immunity by antigenic variation of the variant surface glycoprotein (VSG) coat [6]. Bloodstream forms generate ATP from substrate-level phosphorylation during glycolysis (electronic supplementary material, Note 2). When they attain a sufficient density, oligopeptide quorum sensing [7] triggers the cells to become shorter and fatter stumpy-form trypomastigotes, with $G_0/G_1$ cell-cycle arrest and strongly downregulated transcription and translation [8] (figure 1) (electronic supplementary material, Note 2). Upon uptake by a tsetse fly, the stumpy forms differentiate into procyclic trypomastigotes, which multiply in the midgut. Procyclic forms express different surface proteins—procyclins—and rely on mitochondrial metabolism for ATP (electronic supplementary material, Note 2). Further movement of the parasites through the proventriculus to the salivary glands, differentiation to epimastigotes, a non-obligatory sexual stage, and then conversion to cell-cycle-arrested, VSG-expressing metacyclic trypomastigotes completes the cycle [9].

Leishmanias live in various vertebrates, where they grow inside modified macrophage lysosomes as amastigotes, and within the sandfly digestive system as promastigotes (figure 1). *Trypanosoma cruzi* grows as epimastigotes in reduviid bugs, converting to transmissible metacyclic forms. Within mammals, amastigotes divide in the cytosol of various cell types, while non-dividing extracellular trypomastigotes are responsible for transmission (electronic supplementary material, Note 3).

open reading frames are arranged in a head-to-tail fashion, sometimes with tandem repeats, and with only occasional changes in direction (figure 2). Most RNA polymerase II transcription initiates in DNA stretches of several kilobases that separate gene arrays orientated in opposite, diverging directions (figure 2(1)). Transcription units can be over 100 kb long. Although the gene order is partially evolutionarily conserved [12], individual transcription units contain genes that are unrelated with regard to both mRNA expression and the functions of the encoded proteins. Termination occurs either where two transcription units converge (figure 2(3)) or at regions transcribed by other polymerases (e.g. tRNA and rRNA genes). The primary transcript is processed into individual mRNAs by *trans*-splicing of a capped 'spliced leader' (*SL*) [13], and by polyadenylation [14] (figure 2(4)). *T. brucei* has only two verified *cis*-spliced genes, and none has been confirmed for other kinetoplastids (electronic supplementary material, Note 4). To compensate for the lack of 'activatable' polymerase II transcription, genes encoding abundant mRNAs are often present in multiple copies (illustrated in cyan in figure 2) (electronic supplementary material, Note 5).

In *T. brucei*, the telomeres carry long arrays of *VSG* genes and pseudogenes [15]. The active *VSG* gene is transcribed by RNA polymerase I from one of several alternative expression sites, most of which contain other genes upstream of the *VSG* gene. Antigenic switching can occur either by a transcriptional switch to use of an alternative expression site and promoter, or by recombination or gene conversion events that change the *VSG* sequence within the currently active expression site [6].

# 2. Genome organization and transcription in kinetoplastids

## 2.1. Genome organization

All Kinetoplastea genomes that have been studied to date have similar overall structures [10,11]. Multiple intron-less

## 2.2. Transcription initiation by RNA polymerase II

Trypanosome RNA polymerase II has 12 subunits, which are homologues of those from other eukaryotes. The basal polymerase II transcription factors are more divergent from those

**Figure 2.** Gene expression in kinetoplastids. This diagram is modified from [11]. The thick line at the top represents the genome, and each coloured block is represented in a mature mRNA. The corresponding mRNAs are shown with the coding regions thicker than the untranslated regions. The numbered steps are described in the text. 1: Modified histones in an RNA polymerase II initiation region. 2: RNA polymerase II elongation. 3: RNA polymerase II termination. 4: Endo-nuclease cleavage of precursor (postulated). 5: *Trans*-splicing and polyadenylation. 6: Incompletely processed mRNAs can be degraded by the exosome. 7: Export of a completed mRNA, with bound poly(A) binding protein (PABP), exon junction complex (EJC) and nuclear cap-binding complex (CBC). (Note that we do not know which, if any, splice junctions are bound by the EJC, and also that mRNA export can commence before the 3′-end is complete.) 8: Emergence of a mature mRNA including proteins on the coding region (a) and a specific, stabilizing protein (b) bound to the 3′-untranslated region (3′-UTR). 9: Binding by a silencing or aggregating RNA-binding protein (c) and condensation into granules. 10: Binding of EIF4E, EIF4G and EIF4A, and translation. 11: Protein (b) is replaced by a destabilizing RNA-binding protein (d) and deadenylation starts. 12: Decapping by ALPH1. 13: Degradation by XRNA and the exosome. 14: Rapid decay pathway—immediate decapping promoted by protein (e). Many of the proteins involved are listed in electronic supplementary material, table S1.

of opisthokonts, and include both conserved and novel proteins [16,17]. They include TBP (originally called TRF4) [18], TFIIB [19], the small TFIIA subunit and SNAPc [16], TFIIH [20], Mediator complex [21], a PAF complex [22] and two kinetoplastid-specific proteins that may form the equivalent of TFIIF [17,19]. The general transcription factors are required for *SLRNA* transcription *in vitro*, and it seems very likely that they are needed for polycistronic transcription of protein-coding genes too, although in most cases, this has not been tested. Chromatin immunoprecipitation (ChIP-Seq) experiments should resolve this issue.

In opisthokonts, the repetitive C-terminal domain (CTD) of the largest subunit of RNA polymerase II is phosphorylated; and in humans, this is effected by the CDK7 kinase component of TFIIH [23]. Although the *T. brucei* largest pol II subunit is phosphorylated [24] and the C-terminus is required for RNA synthesis [25,26], it lacks repeats [27]. *T. brucei* TFIIH also has no associated kinase and did not co-purify with RNA polymerase II [28] (electronic supplementary material, Note 6).

Transcription start sites (figure 2(1)) for mRNAs were mapped by purification and sequencing of 5′-triphosphate RNAs. Most start sites are scattered throughout regions that separate divergent transcription units, with clear overlaps between the initiation sites for each direction; some start sites instead allow for re-initiation without a direction change [29,30]. Initiation regions lack open reading frames, have open chromatin [30] and are enriched in H4K10ac, H3K4ac, the histone variants H2AZ, H2BV, the bromodomain factor BDF3 [31,32] and RNA–DNA hybrids [33]. Although several

results indicate that polymerase II initiation can be stimulated merely by the existence of open chromatin [34–36], initiation is favoured by GT-rich sequences [30] (electronic supplementary material, Note 7). It is not known how these sequences are recognized by chromatin modifiers. Only two, rather minor differences in histone variant distribution were observed between life cycle stages [31]. The significance of the differences is unclear, since the parasites used had been cultured separately for many years. It would be useful to find out whether differences are also seen for trypanosomes that have recently differentiated. Results from nuclear run-on experiments (e.g. [37]), as well as results from modelling [38], indicate that transcription elongation (figure 2(2)) is usually constitutive. New results from the sequencing of 5-ethynyl uridine-labelled RNA should provide much more detail about transcription initiation and elongation [39].

The 'retroposon hotspot' (RHS) proteins of *T. brucei* [40] fall into six groups, of which all but RHS6 can be cross-linked to mRNA [41]. By ChIP-Seq, RHS2, RHS4 and RHS6 all colocalize with RNA polymerase II, and RHS6 proteins are directly associated with chromatin [39]. RHS4 and RHS6 proteins are also associated with several other proteins implicated in transcription, whereas RHS2 associates with ribosomes and mRNA-binding proteins [39]. Depletion of any of these proteins resulted in a decrease in transcription, as judged by 5-ethynyl uridine incorporation [39]. These proteins are present only in *Trypanosoma*; it is not known whether equivalent proteins exist in other kinetoplastids.

The 142 nt spliced leader precursor RNA, *SLRNA*, is the only polymerase II transcript that arises from genes with

discrete promoters and terminators [42]. The high *SLRNA* synthesis rate required for *trans*-splicing is attained in part through the presence of about 100 *SLRNA* gene copies arranged as tandem repeats [43]. In *T. cruzi*, this arrangement results in concentration of RNA polymerase II at a specific position within the nucleus [44]. Although *T. cruzi* is at least diploid, only one location is usually seen, perhaps suggesting contact between homologous chromosomes.

Despite the high gene copy number initiation of *SLRNA* transcription has to be much more efficient than at the divergence regions: in bloodstream form *T. brucei*, every *SLRNA* gene needs to make about 140 mRNAs per hour, in contrast to protein-coding genes which are probably transcribed only once or twice per hour [45]. Nucleosomes are strongly phased over the *T. brucei SLRNA* genes, with peaks at the start and termination sites [46]. H2AZ is present, but at a lower level than on the polycistronic unit divergence regions [30]. *SLRNA* promoter recognition and transcription can be studied in an *in vitro* system, and were shown to require a specific initiation complex which contains TBP, distant homologues of human SNAP50 and SNAP43, and a unique component [16,42]. Endoplasmic reticulum stress triggers phosphorylation of TBP and silencing of *SLRNA* transcription, ultimately leading to programmed cell death [47,48].

## 2.3. Polymerase II termination

In opisthokonts, RNA polymerase II termination is linked to polyadenylation. Poly(A) site cleavage seems to suffice for termination, but termination efficiency is enhanced by the nuclear $5' \rightarrow 3'$ exoribonuclease Rat1/Xrn2, which is thought to digest the remnant pre-RNA precursor until it catches up with the polymerase [49]. This polyadenylation-linked 'torpedo' model is not possible in an organism with polycistronic transcription.

Kinetoplastid RNA polymerase II transcription terminates at convergence regions (figure 2(3)), just upstream of tRNA genes [31,35] or pol I-transcribed genes (electronic supplementary material, Note 8), and upstream of the *SLRNA* array [50]. *T. brucei* polymerase II termination regions are enriched with variant histones H3V and H4V [31,50] and cohesin [15]. Histones H3V and H4V are also enriched near telomeres, which have compacted chromatin and are not transcribed by polymerase II [15].

In addition to the two variant histones, a specific modification called base J (beta-D-glucosyl-hydroxymethyluracil) [51] is involved in termination, but the details are species specific. Base J plays a role in termination in bloodstream-form *T. brucei* [50,52,53] and *Leishmania* [54–56] but not in procyclic *T. brucei*, where it is absent [57]. In *T. cruzi*, base J may influence initiation as well [58].

The detailed mechanism of polymerase II termination has not been investigated. Intriguingly, the essential nuclear RNA-binding protein RPB33 binds preferentially to RNAs from regions where polymerase II transcription units converge, and its loss leads to a considerable increase in RNAs that originate from the wrong strand, as well as retroposon and repeat RNAs [59]. These results implicate RBP33 in termination. How RBP33 is recruited to nascent RNAs from termination regions, or targets them for destruction, is not known (electronic supplementary material, Note 9). Kinetoplastids do have a nuclear Rat1/Xrn2 homologue, XRND, which is essential [60], but its detailed function has not been investigated.

In *Leishmania tarentolae*, *SLRNA* transcription terminates just downstream of the mature 3'-end, within an oligo d(T) stretch that is important for, but not the sole determinant of, termination [61]. Oligo d(T) tracts are found downstream of *SLRNA* genes in other trypanosomatids as well. After termination, the mature *SLRNA* is trimmed, probably by a $3' \rightarrow 5'$ exoribonuclease that is also required for processing other small RNAs [62]. The mechanism of *SLRNA* transcription termination is clearly different from that for mRNAs, since oligo d(T) tracts are scattered throughout the DNA that specifies mRNA 3'-untranslated regions (UTRs) and intergenic regions.

## 2.4. RNA polymerase I transcription of protein-coding genes

VSG expression sites and the procyclin loci are transcribed by RNA polymerase I [63], with expression in bloodstream and procyclic forms, respectively [64]. This review concentrates on regulation of genes that are transcribed by RNA polymerase II so evidence concerning VSG and procyclin expression will be summarized only extremely briefly.

*VSG*, procyclin and *rRNA* genes have discrete promoter sequences with clear—but completely different—consensus sequences. So far, no transcription factor that binds just one of these promoter types has been found [65]. Use of polymerase I to make *VSG* mRNA is necessary in order to obtain adequate mRNA levels from the single active *VSG* gene [66]: transcription of a gene from a polymerase I promoter can give at least 10 times more RNA than read-through transcription by RNA polymerase II [64]. The nature of the associated histones may affect this, since the transcription efficiency is affected by the location of the promoter in the genome [30,64,67–70]. The mechanisms of *VSG* and procyclin gene promoter developmental regulation are not understood, but chromosomal location and epigenetic effects are implicated.

The complex mechanism by which all but one *VSG* expression site is suppressed in bloodstream forms has been studied in considerable detail. Very briefly, chromatin structure [15,67,71], sub-nuclear localization [72] and transcription elongation [73], rather than the precise promoter sequence [74], appear to be important. For further information and references to the very extensive literature concerning VSG transcription and antigenic variation, see [15,33,75,76].

# 3. mRNA processing and export

## 3.1. The 5' cap and *SLRNA* processing

Mature mRNAs acquire their cap via *trans*-splicing, and all have the same 39 nt at the 5'-end (electronic supplementary material, Note 10). The structure is $m^7G(5')ppp(5')m_2^6$-AmpAmpCmpm$^3$Ump, with 2'-O ribose methylation of the first four transcribed residues [77,78], as well as $N^6,N^6,2'$-O-trimethyladenosine and 3,2'-O-dimethyluridine [79]. Although the guanylyl transferase [80] and enzymes responsible for ribose methylation [81,82] are known, those responsible for base methylation are not [83]. Cap methylation is required for *trans*-splicing [84], and the ribose methylation is needed for efficient translation [82]. *SLRNA* is also pseudouridylated, although this is not required for function [85].

Capping and methylation of *SLRNA* are co-transcriptional [86], as for polymerase II transcripts in other eukaryotes. In

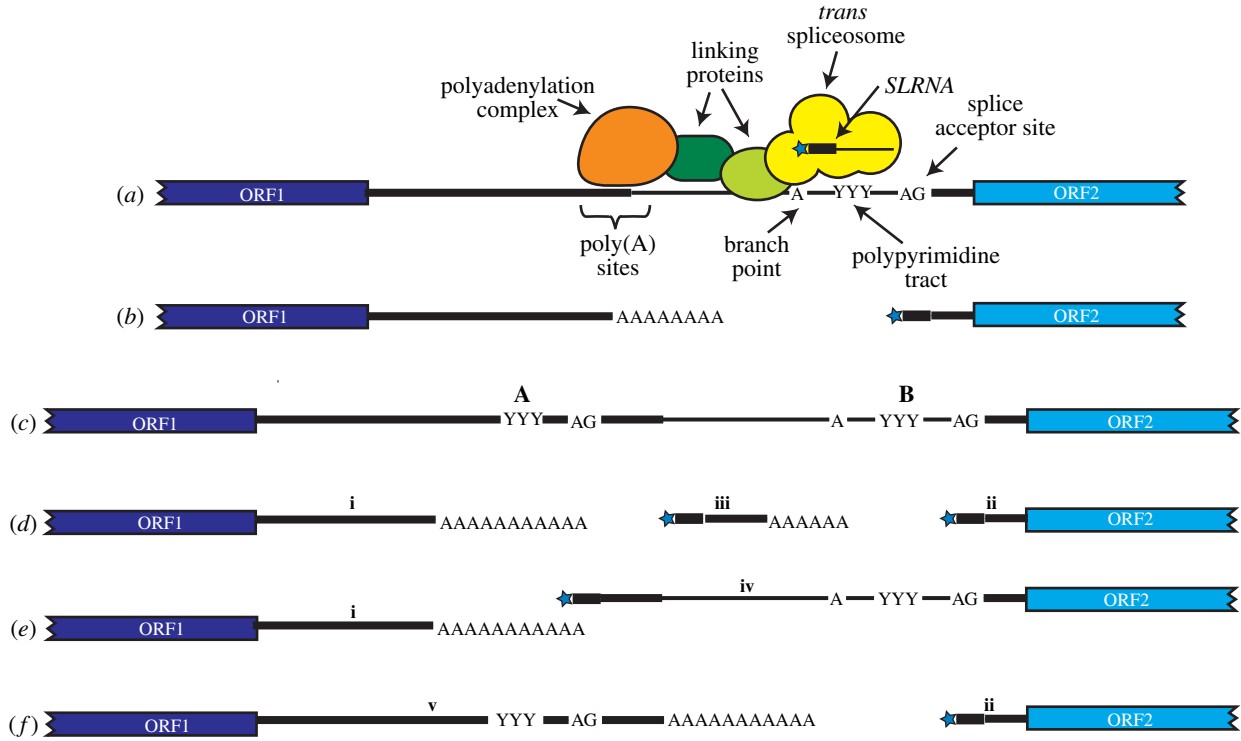

**Figure 3.** mRNA processing. (*a*) This shows the region between two different open reading frames, ORF1 and ORF2. The distance between the polypyrimidine tract and polyadenylation site in *T. brucei* is about 100 nt. The nature of the interaction between the polyadenylation and splicing complexes is unknown; hypothetical linking proteins are shown but the interaction might instead be direct. (*b*) The simplest possibility for processing the primary transcript in (*a*), when only a single splicing signal is present. (*c*) This is a precursor with two possible splicing signals, labelled as A and B. If neither signal is used, the mRNA will contain more than one open reading frame, and only protein (1) will be produced. (*d*) If both sites from precursor (*c*) are used, the RNA from ORF1 (i) has a short 3′-UTR, RNA (2) has a short 5′-UTR (ii) and there will be an additional processed RNA with no open reading frame from intergenic region (iii). This pattern has been documented in detail for the procyclin [101,102] mRNAs but is doubtless seen in many more [29]. (*e*) If only site A of precursor (*c*) is used, the RNA for ORF2 will have a long 5′-UTR (iv) that includes splicing signal (*b*). (*f*) If only the distal site B f precursor (*c*) is used, the ORF1 mRNA will have a long 3′-UTR (v) which includes polypyrimidine tract (*a*).

opisthokonts, the capping enzyme is associated with RNA polymerase II via interactions with both the body of the enzyme and the CTD. Presumably, the capping machinery is also associated with the *SLRNA* transcription complex in trypanosomes, but how this occurs is not known. If indeed capping is specific to the *SLRNA*, involvement of promoter-specific transcription factors seems possible. Depletion of the kinase CRK9, which is required for CTD phosphorylation, impaired cap methylation, but did not impair either capping itself or (surprisingly) *SLRNA* transcription [87]. Interestingly, depletion of a PRP19 complex component, which inhibits splicing (see below), resulted in similar accumulation of unmethylated *SLRNA* [88]. These observations might indicate feedback control of *SLRNA* cap methylation. However, they might also reflect growth arrest, or a first step in *SLRNA* disposal: if assembly of Sm-containing particles is inhibited by depletion of an Sm protein, *SLRNA* cannot be properly assembled as a small nuclear ribonucleoprotein particle (snRNP). Instead, unmethylated *SLRNA* accumulates, first in the nucleus but then in the cytoplasm [89].

## 3.2. Splicing and polyadenylation: basal machineries and signals

*Trans*-splicing was first discovered in trypanosomes [90,91], but was later found in many other very diverse organisms [92] (electronic supplementary material, Note 11). Trypanosomatid *trans*-splicing is mechanistically similar to *cis*-splicing,

with U2, U4, U5 and U6 snRNPs; but the *SLRNA* snRNP takes the place of the U1 snRNP. The components of the spliceosomal snRNPs are broadly similar to those of other eukaryotes, with Sm core proteins [93,94] and specific components [13,95,96]. The U snRNPs undergo SMN-dependent assembly in the nucleoplasm [97,98]. A minimal U1 snRNP is implicated in *cis*-splicing of two transcripts [99]; whether it is also involved in *trans*-splicing is unclear [94,100]. The PRP19 complex [88] is required for both *cis*- and *trans*-splicing [94].

Like 3′ *cis* splice acceptor sites in other organisms, *trans*-splicing sites are usually preceded by a polypyrimidine tract (figure 3*a*) [29,103]. There is a preference for U over C in *T. brucei* but not in *Leishmania major* [104,105]. Longer polypyrimidine tracts give more efficient processing [69,106] and the first downstream AG is the preferred 3′ acceptor [29,107]. During *trans*-splicing, a branched Y-structure intermediate is formed when the *SLRNA* intron is 2′-5′ joined to an A residue in the intergenic region upstream of the polypyrimidine tract: this is equivalent to the lariat that is formed during *cis*-splicing. There is no consensus branch point sequence; evidence from a few genes suggests use of the A residues that are nearest to the polypyrimidine tract (figure 3*a*) [107–109] (electronic supplementary material, Note 12).

Two genes encode potential poly(A) polymerases in *T. brucei*. One mRNA is *cis* spliced, and its product is implicated in small nucleolar RNA (snoRNA) processing [110]. The other, which has no intron, encodes the major mRNA poly(A) polymerase [111]. The polyadenylation complex is conventional apart from two subunits that do not have

royalsocietypublishing.org/journal/rsob    Open Biol. 9: 190072

obvious homologues in unrelated eukaryotes [111]. Polyadenylation site specification is, by contrast, not conventional: instead of there being a dedicated polyadenylation signal, the processing apparatus 'measures' a species-specific distance upstream from the active polypyrimidine tract [14,29,105,112] (figures 2(5) and 3a). The only distinguishing feature of the cleavage site itself is a preference for strings of A residues (electronic supplementary material, Note 13). The 'measuring' mechanism usually results in the use of a cluster of alternative polyadenylation sites. The nature of the link between the splicing and polyadenylation machineries (figure 3a) is not known. Rather oddly, with the exception of one study of AG acceptor dinucleotides [107], there have so far been no studies of trypanosomatid mRNA processing that quantitatively examined the effects of splicing signal mutations on the abundances and processing patterns of both the trans-spliced and upstream polyadenylated mRNAs.

The benzoxaborole acoziborole is undergoing clinical trials for sleeping sickness treatment [113], and other benzoxaboroles are under consideration for other kinetoplastid diseases [114]. The first detectable effect after treatment of T. brucei with the benzoxaborole AN7973 is an increase in mRNA precursors and a decrease in the Y-structure intermediate [3], suggesting that this compound targets mRNA processing. CPSF3 (also called CPSF73) is the enzyme that cleaves the substrate at the polyadenylation site. Overexpression of CPSF3 gives moderate resistance to AN7973 [3] and to acoziborole [4]. Both compounds can be docked into the CPSF3 active site [3,4]. These results suggest that CPSF3 is a primary (though possibly not the only) target of these particular benzoxaboroles (electronic supplementary material, Note 14).

In opisthokonts and plants, there is cross-talk between splicing, RNA polymerase II elongation complexes [115] and chromatin [116] (electronic supplementary material, Note 15). Strict coupling between polymerase II and cis-splicing is based on the requirement for a 5′ cap on the 5′ exon. In kinetoplastids, the SLRNA is equivalent to the 5′ exon. mRNA precursors provide only the 3′ acceptor site, so the polymerase that makes them appears not to influence splicing—hence the ready production of mRNAs using RNA polymerase I or bacteriophage polymerases [117,118]. Kinetoplastid mRNA processing is, as in other eukaryotes, co-transcriptional [119], but interactions between the splicing machinery and the polymerase—if they are present at all—apparently do not influence splicing kinetics [118]. By the time processing occurs, the polymerase is probably roughly 1 kb downstream [38,45]. Nucleosome positioning is phased near or over processing sites, but the patterns in T. brucei and Leishmania are different [46,120], so might be influenced by nucleotide composition rather than by processing activity (electronic supplementary material, Note 16).

### 3.3. Splicing and polyadenylation: use of alternative processing sites

Trypanosomatid 3′-UTRs and intergenic regions are riddled with low-complexity sequences, which often contain polypyrimidine tracts. This can have functional consequences—for example, the use of alternative splice acceptor sites can result in different encoded proteins, and inclusion or exclusion of subcellular localization signals [121,122]. The sequences of UTRs can affect translation and mRNA turnover (see later). Figure 3c illustrates a relatively simply case in which there are two possible splicing signals, A and B. In figure 3d, both are used, making polyadenylation of the RNA with ORF1 independent of splicing of the RNA with ORF2. In figure 3e, only the upstream site is used, giving ORF2 mRNA an extended 5′-UTR (iv), so if there is an AUG and open reading frame between A and B, ORF2 will be very inefficiently translated. In figure 3f, only the downstream signal is used, and 3′-UTR (v) retains a splicing signal A, which could result in bound splicing factors being exported with the mRNA out of the nucleus. 3′-UTR (v) may also contain regulatory sequences that are absent in 3′-UTR (i). The presence of alternative splicing signals in 3′-UTRs may explain why depletion of the basal factors U2AF35, U2AF65 and SF1 affects mRNA stability as well as splicing [123] (electronic supplementary material, Note 17). Why some processing signals are preferred over others is not evident. Since alternative processing can affect mRNA abundance, the amounts of different alternative mature mRNAs do not necessarily indicate which processing sites are preferred.

Limited results from reporter experiments suggest that when there are several polypyrimidine tracts, the one that is closest to the downstream initiator AUG is usually preferred [124,125]. This might be explained by binding of specific protein factors in the 5′-UTR, at the start of the coding region or upstream of the polypyrimidine tract. As in other eukaryotes, RNA polymerase II occupancy is higher over coding regions [46]. This might mean that elongation is slowed relative to intergenic regions, which in turn might favour use of coding-region-proximal splice sites.

### 3.4. Quality control of RNA processing

If transcription, processing and mRNA export were all constitutive, the levels of mRNAs would depend solely on degradation of the processed mRNAs [38,126]. However, many mRNAs are much less abundant than expected from their half-lives [38,126], with a strong bias against longer mRNAs. Mathematical modelling showed that this could be explained by assuming that mRNA splicing competes with a nuclear quality control mechanism that attacks precursors stochastically (figure 2(4)) [38,126]. A direct test, however, failed to confirm this hypothesis [127], so the anomalous low abundance of longer mRNAs remains unexplained (electronic supplementary material, Note 18). Nevertheless, degradation of precursors and discarded intergenic regions clearly occurs [45].

The RNA exosome is a 3′→5′ exoribonuclease complex. A core structure that is responsible for substrate recognition and RNA unwinding [128] is associated with the 3′→5′ exonuclease Rrp6 and a combined 3′→5′ exonuclease and endonuclease, Rrp44 (electronic supplementary material, Note 19). In trypanosomes, as in other eukaryotes, the principal role of the exosome is probably in the nucleus, especially processing of stable RNAs such as rRNA [129,130]. The trypanosome exosome is stably associated only with a single exoribonuclease, RRP6. Kinetoplastids have RRP44, but no association of it with the exosome has been detected (electronic supplementary material, Note 20). When trans-splicing is impaired, less efficient processing sites can be skipped, yielding trans-spliced and polyadenylated mRNAs that contain more than one open reading frame, joined by the intergenic

region (figure 3c; electronic supplementary material, Note 21). These can accumulate and escape into the cytoplasm [131]. Interestingly, depletion of the exosome resulted in accumulation of such mRNAs [130] (electronic supplementary material, Note 22). This suggests that such partially processed mRNAs may normally be targeted for destruction by the exosome (figure 2(6)). This would require endonuclease cleavage prior to 3′→5′ digestion—could RRP44 be responsible?

Many questions remain about kinetoplastid *trans*-splicing. Why are some polypyrimidine tracts favoured for splicing, while others are unused (figure 3f)? How do splicing kinetics affect the abundance of the upstream mRNA? And how are partially processed mRNAs recognized by the nuclear quality control machinery? Some of these issues may be resolved by studying the roles of potential splicing regulators.

## 3.5. Splicing and polyadenylation: possible specific regulators

Metazoan proteins with serine–arginine-rich (SR) domains are sequence-specific splicing regulators which mainly bind immediately downstream of exon junctions [132–134]. SR proteins are important in defining exon–intron boundaries, but may sometimes also influence polymerase II elongation. They can accompany mature mRNAs into the cytoplasm, but are mostly not associated with translating mRNAs. In addition to the basal splicing factor U2AF65, at least five kinetoplastid proteins combine one or more N-terminal RNA recognition motifs (RRMs) with C-terminal SR domains: TRRM1 (also called RRM1), RBSR1, RBSR2, RBSR3 and TSR1. All (except RBSR2, which has not been tested) are in the nucleus, although green fluorescent protein-tagged RBSR3 is in the cytoplasm as well [135].

TRRM1 is mostly in the nucleus [136] and is essential [137,138]. Its association with PTB2/DRBD4 (see below) suggests a role in splicing [138], although most splice sites were unchanged after TRRM1 depletion. The mRNAs that changed in abundance after TRRM1 depletion were mostly not the same as those that were bound to it. Longer, less abundant RNAs were more bound than shorter ones; but as noted above, length and abundance are not independent parameters. Interestingly, TRRM1 associated with some RHS proteins and nucleosome abundance increased in some regions upon TRRM1 loss [138]; whether this was a direct effect or secondary to growth inhibition is not yet clear.

RBSR1 is absent in *Leishmania*. In *T. cruzi* [139], it is associated with snoRNAs and snRNAs. A pull-down suggested association with TRRM1, RBSR2 and four other abundant RNA-binding proteins: UBP1, UBP2, ALBA3 and ALBA4. TSR1 interacts with TSR1IP, a highly polar protein [140]. Both are localized primarily in nuclear speckles. TSR1IP and TSR1 purifications contain TRRM1 and three other proteins with RNA-binding domains (electronic supplementary material, Note 23). Depletion of either resulted in an overall decrease in splicing, but microarray results revealed only minor changes in mature mRNA abundances [140].

In addition to SR proteins, several other RRM-containing proteins are localized primarily in the nucleus and have been implicated in splicing. DRBD3 (PTB1) and DRBD4 (PTB2) (electronic supplementary material, Note 24) are both essential, and were suggested as homologues of the polypyrimidine tract binding proteins of opisthokonts [141,142]. Depletion of either caused decreases in both *SLRNA* and the Y-structure intermediate. Many mRNAs showed altered abundances, but only some of the mRNA decreases could be linked to impaired splicing [142]. The authors suggested that mRNAs with C-rich polypyrimidine tracts were more strongly affected and indeed, a later study of binding sites identified a preference of DRBD3 for a C-rich motif [143].

HNRNPH/F is a hundred times more abundant in bloodstream forms than procyclic forms, but its depletion inhibited splicing in both forms. A putative binding motif was found in 5′-UTRs of some mRNAs whose splicing was particularly impaired after HNRNPH/F depletion [144]. Microarray analyses also suggested that HNRNPH/F has additional roles in regulating mRNA stability; these will be discussed later.

Many splicing factor studies so far were limited by microarray technology. Our understanding would be considerably increased by detailed mapping of RNA-binding sites. If a factor is bound near a *trans*-splice acceptor site, it is likely to affect processing, while binding elsewhere within the mature mRNA could influence mRNA export, translation or decay. We do not yet know how many of the proteins discussed above are exported bound to mRNAs (electronic supplementary material, Notes 17 and 25).

## 3.6. Export of mRNAs to the cytoplasm

The universally conserved system for mRNA export from the nucleus (figure 2(7, 8)) consists of homologues of yeast Mex67, Yra1 Sub2 and Mtr2 [145]. As expected, RNAi targeting *T. brucei* MEX67 [146], or its interaction partners MTR2 and importin 1 [147], causes accumulation of poly(A)+ RNAs in the nucleus. Unusually, kinetoplastid MEX67 has an N-terminal zinc finger domain which is essential for function and cell survival [147]. In opisthokonts, mRNA export factors are recruited to mRNAs via the SAGA, THO and TREX complexes. In most other eukaryotes, including kinetoplastids, these complexes are missing [145] and the mechanisms by which mature mRNAs are recognized for export are not known.

An *in situ* hybridization study demonstrated that in trypanosomes, mRNA export can commence before the 3′-end has been synthesized [148]—in other words, it can be co-transcriptional. When *trans*-splicing is inhibited, granules containing mRNAs accumulate on the cytosolic side of the nuclear pore [131]. The authors examined a very long mRNA—22 kb—and found that after treatment with sinefungin, a 5′-processed version that lacked a 3′-end got stuck in the nuclear pore channel [148]. This indicates that the export machinery recognizes processed 5′-ends of mRNAs, rather than the poly(A) tail. There are three obvious options for this. MEX67 might recognize the *SL* sequence (via the zinc finger?), or it might bind to the nuclear cap-binding complex or the exon junction complex.

The opisthokont nuclear cap-binding complex, CBP20/CBP80, stimulates splicing through interaction with snRNPs, and promotes mRNA export via the TREX complex [149,150]. The trypanosome *SL* cap-binding complex contains the cap-binding component CBP20, and three additional proteins that are unrelated to CBP80 [151] (electronic supplementary material, Note 26). Depletion of any CBP inhibits growth and *trans*-splicing. Poly(A) binding proteins bind to poly(A) tails in all eukaryotes examined so far

royalsocietypublishing.org/journal/rsob Open Biol. 9: 190072

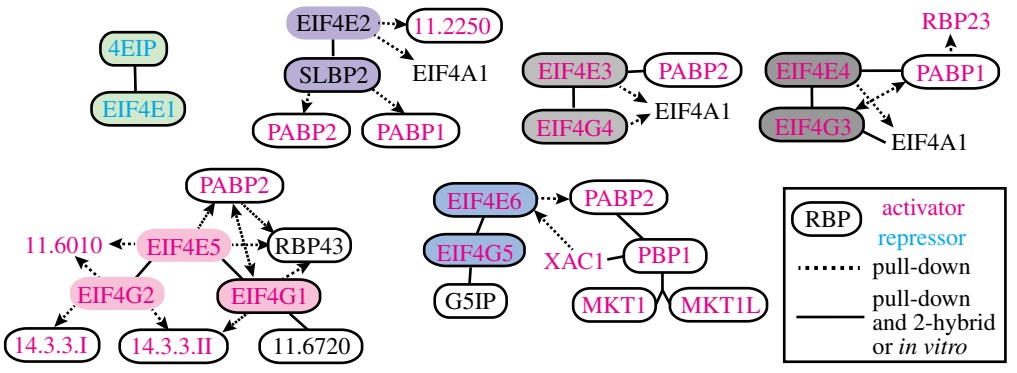

**Figure 4.** Interactions of cap-binding translation initiation complexes in *T. brucei* and *Leishmania*. The key is on the bottom right. The colours of the ovals identify different EIF4E interaction partners. Text colour codes for 'activator' (magenta) and 'repressor' (cyan) indicate whether the protein activates, or represses, gene expression when tethered within the 3′-UTR of a reporter mRNA in bloodstream-form *T. brucei* [41,167] (electronic supplementary material, Note 30). The solid frames indicate that the protein could be UV-cross-linked to poly(A)+ RNA in bloodstream-form *T. brucei* [41]. Gene numbers are from *T. brucei*, with the prefix 'Tb927.' removed. For results that rely on pull-down only, arrows point away from the bait protein. Suggested stage-specific interactions between *Leishmania* EIF4E1 and both EIF3 and EIF4G3 [168] are not shown. Binding activities of *Leishmania* EIF4E4 to modified caps are reported in [169,170]. Alternative names are G1-IP2 for RBP43, and G1-IP for Tb927.11.6720. References for pull-down and interaction results are as follows: *T. brucei* PABPs [155]; *Leishmania* and *T. brucei* EIF4E1 [168,171]; *T. brucei* EIF4E2 [172]; *Leishmania* EIF4G3 [173]; *T. brucei* and *Leishmania* EIF4E3 and EIF4E4 [168,174]; *T. brucei* EIF4E5 [175]; *T. brucei* EIF4E6 [176]. The tethering results for EIF4E5 and EIF4E6 are unpublished (L. Melo do Nascimento and C. Clayton 2019, unpublished data); note that the activities of these proteins when bound to the 5′ cap may not be the same as those seen in the tethering assay.

[152]. *T. brucei* has two poly(A) binding proteins, PABP1 and PABP2 [153]; the association of PABP2 with CBPs [154,155] suggests that PABP2 may accompany new mRNAs from the nucleus to the cytoplasm.

The exon junction complex (EJC) [156,157] seems the most likely candidate for export recognition. In mammals, the EJC component eIF4AIII (a DEAD-box helicase) is recruited by splicing factor CWC22 and deposited 20–24 nt upstream of splice junctions [156–158]. A stable dimer of Y14 and Magoh/RBM3 [158] then binds to eIF4AIII (electronic supplementary material, Note 27). Various additional proteins interact with the EJC and facilitate mRNA export [157]. *T. brucei* Y14 is essential for normal growth and forms a stable complex with the Magoh homologue [159]. *T. brucei* also has CWC22 and a nuclear eIF4A-like helicase that is probably equivalent to eIF4AIII [160] (electronic supplementary material, Note 28). Although no association of the *T. brucei* eIF4AIII with Y14 has been detected [159], this might be because the interaction is quite transient. We do not currently know which splice junctions—if any—are bound by the kinetoplastid EJC. However, if the EJC is deposited on the *SL* at the 5′-UTR of each mRNA after *trans*-splicing (as shown in figure 2(7, 8)), it would be an excellent marker for newly processed mRNAs.

# 4. Translation

## 4.1. Cap-binding translation initiation complexes

Eukaryotic translation initiation usually commences with binding of the 5′ cap by initiation factor eIF4E (figure 2(10)). In most (but not all) organisms [161], eIF4E recruits eIF4G, which in turn recruits the helicase eiF4A1. Meanwhile, various other factors, including eIF2 and eIF3, are included during the assembly of the 43S complex, which contains a charged methionyl tRNA bound to the 40S ribosomal subunit. Phosphorylation of the alpha subunit of eIF2 prevents 43S complex assembly. Interaction of cap-associated eIF4G with

eIF3 recruits the 43S complex to mRNA, forming a 48S complex. This then scans to the initiation codon, where it is joined by a 60S subunit [162–164]. An interaction between eIF4G and poly(A) binding protein can, at least transiently, circularize the mRNA, promoting re-initiation and protecting against exonucleolytic degradation [148,165,166] (electronic supplementary material, Note 29).

*T. brucei* PABP1 and PABP2 co-sediment with polysomes under normal conditions and each is essential, implying different functions [154]. Leishmanias have an additional PABP paralogue, PABP3, which is closer to PABP1 than to PABP2 [154].

Kinetoplastids have an unusually large repertoire of eIF4Es and eIF4Gs (figure 4), which were reviewed in detail in [177]. EIF4E3, EIF4E4, EIF4E5 and EIF4E6 appear to be 'classical' EIF4Es, since each interacts with at least one EIF4G (figure 4). They also activate expression when tethered to the 3′-end of a reporter mRNA, suggesting that they are likely to be active in translation initiation. (For details of this assay see electronic supplementary material, Note 30.) Both EIF4E3 and EIF4E4 have unusual N-terminal extensions, and are about 100 times more abundant than EIF4E2 and EIF4E1 [174] so would suffice for translation of all mRNAs [178]. RNAi results, combined with failed knock-out attempts, indicate that the EIF4E3- and EIF4E4-based complexes are essential for general translation and cell survival [174]. Interestingly, the N-terminal extension of *Leishmania* EIF4E4 associates directly with PABP1 [179], while the C-terminal conserved domain is responsible for recruiting EIF4G3. Whether EIF4G3 also interacts directly with PABP1 is controversial [168,179,180]. EIF4E3–EIF4G4 can associate with both PABPs [155]. It is not yet known whether these complexes target different mRNAs. *Leishmania* EIF4E4 may be active only in promastigotes: its expression is downregulated in axenic amastigotes, where no EIF4E4 cap binding is detected [168,173].

EIF4E5 has two alternative EIF4G partners, EIF4G1 and EIF4G2 (figure 4) [175]; of these, EIF4G1 interacts with an RRM-domain protein (RBP43), with a protein that may

royalsocietypublishing.org/journal/rsob    Open Biol. 9: 190072

modify or repair the 5′ cap [175], and with PABP2 [154,155]. *EIF4E5* RNAi in procyclics caused a motility defect [175]. Quantitative mass spectrometry results suggest that EIF4E6 is at least as abundant as EIF4E3 and EIF4E4, and that EIF4G5 is as abundant as EIF4G4 [181]. EIF4E6 might, therefore, have a constitutive function. RNAi targeting *EIF4E6* in procyclics resulted in flagellar detachment [176]. (See also electronic supplementary material, Note 31.)

Both EIF4E1 and EIF4E2 are, unlike the other EIF4Es, found in the nucleus as well as the cytoplasm and have lower abundances than EIF4E3 and EIF4E4 [174]. Neither co-purifies with an EIF4G. EIF4E2 pairs with a protein that has a double-stranded RNA-binding domain [172]. The functions of EIF4E2 and its partner are not known.

EIF4E1 is the only *T. brucei* EIF4E that suppresses expression when tethered. It interacts with a protein called 4EIP in both *Leishmania* [168] and *T. brucei* [171]. 4EIP binds directly to mRNA [41] and, when tethered, is a suppressor even when EIF4E1 is absent [171]. By contrast, the suppressive activity of EIF4E1 depends on its interaction with 4EIP. The phenotypes of trypanosome depletion mutants suggest that the two proteins may act both together and independently. 4EIP is required to suppress translation in stumpy forms, but EIF4E1 is not; by contrast, procyclic forms require EIF4E1 but not 4EIP [171]. The results suggest that EIF4E1 is either unable to initiate translation independently, or is able to do so only weakly. Binding of 4EIP to EIF4E1 stabilizes EIF4E1, but simultaneously reduces the EIF4E1–cap interaction [182] (electronic supplementary material, Note 32).

In mammalian cells, the first or 'pioneer' round of translation involves eIF4AIII [183] and is initiated by a specialized eIF4G-like protein that interacts with CBP80 [184]. Nothing is known about the pioneer round of translation in kinetoplastids but there is no indication that the nuclear cap-binding complex has any translation-stimulating properties.

## 4.2. Scanning, elongation and termination

Before initiation, the 48S complex has to scan through the 5′-UTR in order to reach the initiation codon. Scanning can be impeded by secondary structures, which are unwound by eIF4A and additional RNA helicases [185–188]. Most kinetoplastid 5′-UTRs are under 100 nt long, and they rarely exceed 250 nt [29,103,189]. In *T. brucei*, both hairpins and 5′-UTRs longer than 200 nt were shown to decrease expression of a reporter protein [190], suggesting that 48S complexes may be lost during scanning. *T. brucei* has 13 predicted DEAD/H-box helicases, of which eight are known to be in the cytosol (electronic supplementary material, table S1). With the exception of EIF4A1 and DHH1, the functions of these proteins cannot readily be assigned by sequence comparisons. Two of them, however, are likely to be involved in translation since they can complement a yeast Ded1p mutant [191].

As noted above in relation to figure 3*e*, if there is a short open reading frame upstream of the principal coding region (uORF), translation may initiate and terminate before the principal initiation codon is reached. As a consequence, ribosome densities on opisthokont mRNAs with uORFs are lower than on mRNAs that lack them [192]. Roughly a tenth of *T. brucei* 5′-UTRs contain uORFs with ribosome occupancy [193,194]. The relevant mRNAs have significantly lower ribosome densities on the main ORF than mRNAs without uORFs [193,194] (electronic supplementary material, Note 33). They also have lower mRNA abundances [193,194], although they are only slightly more unstable [38] (electronic supplementary material, Note 34).

In prokaryotes and eukaryotes, rates of translation elongation are strongly influenced by codon composition, and slow translation correlates with lower mRNA abundance [195]. Variations in tRNA abundance are thought to underlie the influence of codon composition, with strings of codons with low tRNA abundances causing upstream ribosome traffic jams [195]. The cellular abundances of tRNAs are not known for kinetoplastids. Nevertheless, two studies have shown that codon composition influences translation and mRNA abundance: sub-optimal codons tend to result in less RNA and less protein [196,197].

In opisthokonts and plants, a quality control pathway called 'nonsense-mediated decay' ensures removal of mutant mRNAs in which a termination codon is present within the normal open reading frame [198,199]. This requires, minimally, termination factors eRF1 and eRF3 and the interacting nonsense-mediated decay factors Upf1 and Upf2 [199]. Kinetoplastids have homologues of all four proteins, and UPF1 and UPF2 mutually interact. However, RNAi targeting had no reproducible effect either [190,200] and it is not clear whether a classical nonsense-mediated decay pathway is present or not [190] (electronic supplementary material, Note 35).

# 5. mRNA decay

## 5.1. Removal of the poly(A) tail and 3′→5′ degradation

Opisthokont mRNA decay usually commences with deadenylation by the NOT complex (figure 2(11)). The NOT complex has been found in all eukaryotes examined so far. It has at least six subunits, and assembles on a large scaffold protein, NOT1; its activity is inhibited by PABP [201–203] (electronic supplementary material, Note 36). NOT complexes from many species are associated with two 3′→5′ exoribonucleases, but trypanosomes have only one— CAF1 [204,205]. Transcriptome-wide measurements of mRNA half-lives after inhibition of RNA synthesis yielded median half-lives of 12 min in bloodstream forms and 20 min in procyclic forms [38] (electronic supplementary material, Note 37). For the majority of mRNAs, deadenylation by the NOT complex was clearly the most important determinant of half-life [206]. RNAi targeting *CAF1* inhibited overall deadenylation, resulting in a marked increase in the half-lives of most mRNAs [204,207].

A second eukaryotic deadenylation complex consists of the enzyme PAN2 bound to a PAN3 dimer [203,208,209]. Mammalian PAN2/PAN3 can digest PABP-bound poly(A) and is important in trimming relatively long poly(A) tails [210]. In *T. brucei*, RNAi targeting PAN2 stabilizes mRNAs with intermediate half-lives [206] but has no effect on the steady-state poly(A) tail distribution and causes only a marginal delay in overall deadenylation [146].

Animal 'poly(A) ribonuclease' (PARN) enzymes have activities at specific developmental stages. Trypanosome genomes encode three PARNs that appear not to be implicated in constitutive deadenylation [211]. Their roles during differentiation or stress have not been studied.

Many questions remain concerning deadenylation in trypanosomatids. We do not know how tail lengths— which vary up to 200 nt [146,204]—are determined, or how deadenylases are differentially targeted to individual mRNAs. Counterintuitively, stable opisthokont mRNAs with high ribosome densities and optimal codon compositions have very short poly(A) tails [212]; the relationship is not known in trypanosomes.

After RNAi targeting the exosome or deadenylation, the effects on steady-state levels of many mRNAs correlated very poorly with the effects on decay rates [206]. Instead, after loss of CAF1, PAN2 or RRP45, changes in mRNA abundance correlated negatively with mRNA length [213]. Since the same was seen after loss of the transcription elongation factor CTR9 [213], is seems possible that loss of $3' \rightarrow 5'$ decay causes feedback inhibition of transcription, but this has not been verified.

## 5.2. Decapping and degradation of the body of the mRNA

The shortest poly(A) tail length that is seen, in both opisthokonts and trypanosomes, is about 30 nt [204], which is slightly above the 25 nt that is occupied by a single PABP [152]. Since binding of opisthokont PABPs to poly(A) tails is enhanced by cooperative interactions [152], it is thought that this final PABP monomer can be removed relatively easily, after which degradation of the body of the mRNA is triggered. Although loss of PABP will expose the mRNA to the exosome, the major pathway is removal of the mRNA cap by decapping enzymes (figure 2(12)), followed by $5' \rightarrow 3'$ degradation (figure 2(13)). This can start while the mRNA is still being translated [214,215]. In trypanosomes too, deadenylation followed by decapping and $5' \rightarrow 3'$ degradation seems to be the predominant decay mechanism.

mRNA decapping enzymes are usually members of the MutT hydrolase family. The products are 5'-monophosphorylated mRNAs, which are substrates for the $5' \rightarrow 3'$ exoribonuclease Xrn1 [83]. Kinetoplastids have five potential MutT hydrolases but none is implicated in decapping of deadenylated mRNAs. Instead, caps are removed by ALPH1, an ApaH-like phosphatase which removes the cap to leave an mRNA with a 5'-diphosphate [216]. An additional 5'-phosphatase may be required before the trypanosome Xrn1 homologue, XRNA [217], can digest the mRNA. We do not know how ALPH1 is recruited to its mRNA targets.

In addition to ALPH1, *T. brucei* has several enzymes that seem likely to be involved in cap repair or quality control. These include a MutT hydrolase that preferentially acts on unmodified caps [218] (electronic supplementary material, Note 38) and a protein with a guanylyltransferase domain that co-purified with mRNA, EIF4E5 and EIF4G1 [41,175]. Finally, an enzyme with activity similar to that of yeast DcpS was detected [219] but there is no obvious homologue in the *T. brucei* genome.

After decapping, XRNA is probably important for the degradation of most mRNAs [217]; this was illustrated at the single-molecule level for one mRNA [220]. The exosome plays a relatively minor role in mRNA degradation [206], and the roles of two additional $5' \rightarrow 3'$ exoribonucleases [60] are unknown. Depletion of XRNA was lethal, but preferentially stabilized mRNAs with very rapid decay, suggesting

that for these mRNAs, $5' \rightarrow 3'$ degradation (rather than deadenylation) was limiting [217] (figure 2 (14)). Detailed examination of the mRNA encoding EP procyclin in bloodstream forms revealed that, rather unusually, it is degraded from both ends, with contributions from XRNA, CAF1, PAN2 and the exosome [60,146,221,222] (electronic supplementary material, Note 39). The combined results suggest that this and other similarly unstable mRNAs are attacked by the $5' \rightarrow 3'$ degradation machinery without prior deadenylation. Nevertheless, this must start after export is complete, since the degradation rates are determined by sequences in the mRNA 3'-UTRs (figure 2 (14)).

In yeast, the RNA helicase Dhh1 is associated with mRNAs with slow elongation, and activates decapping [223]. Trypanosome DHH1 probably has several different roles in gene expression: it has been found in affinity-purified preparations of NOT complex components and various other proteins implicated in mRNA metabolism. Depletion of DHH1, or over-expression of a catalytically dead mutant, in procyclic forms led to accumulation of bloodstream-form-specific mRNAs [224].

## 5.3. The role of untranslated regions

Decay rates are a major determinant of trypanosomatid mRNA abundance. Codon usage contributes strongly to constitutive decay rates, giving at least a 25-fold range [196,197], but most sequences responsible for regulation in response to environmental change are in 3'-UTRs. Often, regions of at least 100 nt are found to be necessary for regulation. This may be because secondary structure is important, or because binding by several different proteins is required. A median *T. brucei* 3'-UTR of about 400 nt [29,103,189] would have space to bind at least 13 proteins (electronic supplementary material, Note 40).

## 5.4. Environmental adaptation and differentiation

Most studies of the mechanism of gene expression regulation have focused on differences between different life cycle stages or responses to individual developmental triggers (figure 1). Interpretation of results with growing cells is relatively straightforward. By contrast, with growth-arrested stages, it is difficult to distinguish genuine developmental regulation from the effects of stress or arrest *per se*. For example, during formation of stumpy *T. brucei*, there is downregulation of both translation and transcription [225,226], and translation arrest by itself can block mRNA degradation [196]. With the exception of naturally obtained life cycle stages, therefore, all results obtained with growth-arrested cells have to be treated with considerable caution (electronic supplementary material, Note 41).

Researchers have also looked for responses to individual nutrients. In *T. brucei*, changes in availability of the main substrates of energy metabolism in bloodstream and procyclic forms (glucose and proline respectively), and of glycerol in both forms, result in remodelling of metabolism [126,227–234]. Stablization of the *GPEET* procyclin mRNA via its 3'-UTR [227] is likely to be an indirect effect of changes in energy metabolism [228]. The increased expression of transferrin receptor mRNAs in response to iron deprivation [235], feedback inhibition of the synthesis of the prozyme subunit of S-adenosylmethionine decarboxylase [236] and

royalsocietypublishing.org/journal/rsob    Open Biol. 9: 190072

repression of purine transporter NT8 expression by purines [237] are also 3′-UTR mediated (electronic supplementary material, Note 42).

## 5.5. Regulation by RNA-binding proteins

All results so far indicate that control by UTRs is effected exclusively via RNA-binding proteins [238]. Although RNAi is active in *T. brucei* and some *Leishmania* species, many kinetoplastids completely lack the RNAi machinery and its role is probably restricted to suppression of retroposons and viruses [239]. There is no experimental evidence for miRNAs [240] (electronic supplementary material, Note 4).

The behaviour and fate of each mRNA will depend on dynamic, combinatorial interactions between that mRNA and the various proteins that compete for binding (e.g. [241]). An annotated list of *T. brucei* proteins with RNA-binding domains is included in electronic supplementary material, table 1, sheet 1. The domains include approximately 90-residue RRMs, CCCH zinc finger domains (consensus: $CX_8CX_5CX_3H$) and pumilio (Puf) domains. Most RNA-binding proteins are likely to be present in considerable excess over their target mRNAs [178], ensuring that, with low nM dissociation constants [242–244], the majority of potential binding sites in mRNAs will be occupied. The combination of mRNA and proteins is often called a 'messenger ribonucleoprotein particle' (mRNP) (electronic supplementary material, Note 43). A catalogue of all proteins bound directly to mRNA in bloodstream forms (mRNP proteome) is already available [41], and the ability of each protein to activate, or repress, expression when tethered to the 3′-UTR of a reporter RNA has also been assessed [41,167] (electronic supplementary material, Notes 30 and 44). In the following section, I will describe the roles of selected RNA-binding proteins in particular aspects of mRNA control.

## 5.6. Cell-cycle regulation

As in other eukaryotes, the abundances of some kinetoplastid mRNAs and proteins vary during the cell cycle [245]. This has been examined in detail only in *Crithidia fasciculata*, where levels of a few cell-cycle-regulated mRNAs are controlled by two complexes called CSBPI and CSBPII [246]. CSBPI consists of two proteins with CCCH zinc finger domains, CSBPA and CSBPB. The trypanosome homologues, ZC3H39 and ZC3H40, mutually interact, bind to mRNA and repress when tethered [41,247] (electronic supplementary material, Note 45).

The *Crithidia* CSBPII complex contains poly(A) binding protein and two other proteins whose activities are controlled by phosphorylation. CSBPII_45 binds specifically to a 5′-UTR motif that is required for cell-cycle regulation (electronic supplementary material, table S1) [248]. Both *T. brucei* CSBPII homologues cross-link to *T. brucei* mRNA and CSBPII_33 is a repressor when tethered [41,167], but the *Crithidia* regulatory motif is not found in trypanosome cell-cycle-regulated mRNAs [245].

In *T. brucei*, the pumilio domain protein PUF9 binds to, and stabilizes, some mRNAs that are preferentially increased during S-phase [249]. A motif that includes the Puf core recognition motif, UGUA, was required for binding and regulation [249]. Stabilization by PUF9 may depend on the MKT complex, which is described later. PUF9 is unusual, because

Puf-domain proteins usually repress translation and promote mRNA decay [250]. *T. brucei* PUFs 1, 2, 3 and 10 indeed suppress expression when tethered (electronic supplementary material, table S1). Results for *Leishmania* [251] and *T. cruzi* PUF6 indicate that it too is a suppressor; *Leishmania* PUF6 contributes to destruction of stage-regulated mRNAs containing a retroposon sequence [251].

## 5.7. Procyclic-form-specific mRNAs

Several hundred mRNAs are much more abundant and/or better translated in procyclic forms than in bloodstream forms. Some of these, including those encoding EP procyclin, cytosolic phosphoglycerate kinase (PGKB) and various cytochrome subunits, have one or more $U(A)U_6$ motifs in their 3′-UTR. This motif is bound by a repressive, bloodstream-form-specific protein called RBP10 [252] (figure 5). Bloodstream forms with depleted RBP10 show increases in many of the target mRNAs [252,256], and can grow only if transferred to procyclic conditions [252]. Conversely, procyclic forms with forced expression of RBP10 can grow only as bloodstream forms [252]. The mechanism by which RBP10 represses translation and promotes mRNA decay is not known: searches for interaction partners did not reliably reveal interactions with the degradation machinery [252,257] (electronic supplementary material, Note 46). RBP9 [257,258] and ZC3H32 [259] are additional bloodstream-form-specific repressors (figure 5) but so far their roles are unclear (electronic supplementary material, Note 47). In yeast, poor translation initiation correlates to some extent with rapid mRNA decay [260]. Similarly, tethering of RBP10 [252], ZC3H32 [259] or 4EIP [171] causes both translation inhibition and mRNA destruction, but in each case, it is difficult to determine what happens first.

The converse side of regulation for procyclic-form-specific mRNAs is stabilization and/or translation enhancement in the procyclic form. To try to identify the proteins resonsible, affinity purification using the $U(A)U_6$-containing regulatory element from *PGKB* was undertaken. This yielded UBP2, RBP33, RBP42 and DRBD3 [261]. These are constitutively expressed abundant RNA-binding proteins with a preference for U-rich sequences. RBP33 was discussed earlier; the remaining proteins will be discussed later.

ZFPs 1, 2 and 3 (figure 5) mutually interact [262]. ZFP1 expression is upregulated during differentiation and remains high in procyclic forms [263], while ZFP2 [263] and ZFP3 [262] are constitutvely expressed. All three are implicated in differentiation. ZFP3 associates with the EP procyclin 3′-UTR [262,264] and might be implicated in the switch from early to late procyclic forms [264], but it also associates with many constitutively expressed mRNAs [265] (electronic supplementary material, Note 48). Expression of the membrane protein ESAG9 is suppressed in long slender bloodstream forms and procyclic forms, and elevated in stumpy forms. A screen for proteins required for the suppression in bloodstream forms revealed ZFP3, DRBD5 and a repressive RNA-binding protein, REG9.1 [266]. REG9.1 is most abundant in procyclic forms, where it is essential [266]. Its mode of action and direct targets are unknown.

RNAi targeting HNRNPH resulted in a loss of developmental regulation, and its putative binding motif was enriched in the UTRs of affected mRNAs. However, the effects of depletion were often opposite in bloodstream and

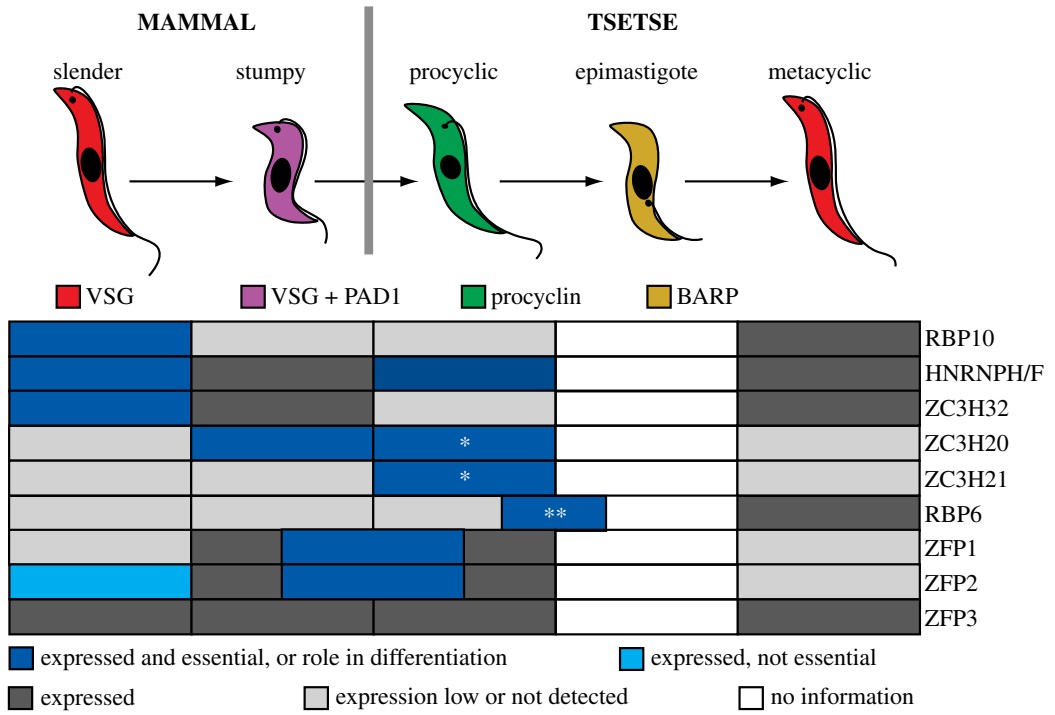

**Figure 5.** RNA-binding proteins that have been implicated in developmental regulation. The figure shows a simplified pathway from the bloodstream form to the metacyclic form. The proteins shown were chosen because they are well characterized. Several more proteins have been shown to be essential for differentiation; these, as well as other RNA-binding proteins that are expressed in a stage-specific manner, but whose roles in differentiation have yet to be defined in detail, are mentioned in the text or can be found in electronic supplementary material, table S1. Regulated expression is derived from proteome data for differentiation of bloodstream forms to procyclic forms [181,253,254] and for RBP6-induced differentiation of procyclic forms to metacyclic forms [255]. Proteome data for epimastigotes are not yet available. *RNAi targeting both simultaneously is lethal. **RBP6 expression triggers differentiation to epimastigotes, and the mRNA is present in proventricular and salivary gland parasites, but it is not known whether RBP6 is essential for differentiation. References for the phenotypic results are in the text. 'Essential' usually means that RNAi was detrimental; the absence of an RNAi effect does not necessarily mean that the protein is not required.

procyclic forms. For example, the effect of HNRNPH in *stabilizing* an amino acid transporter (*AATP11*) mRNA in procyclic forms depended on the presence of the binding motif in the 3′-UTR [144]; but the same mRNA was *increased* after HNRNPF/H depletion in bloodstream forms, suggesting that HNRNPH was acting as a *repressor* [144]. It is possible that the activity of HNRNPH is changed by post-translational modification or protein–protein interactions; alternatively, some effects might be secondary. Tethering results for full-length HNRNPH, which might help to resolve this, are not available.

ZC3H20, ZC3H21 and ZC3H22 share a region containing two CCCH zinc finger domains; their mRNAs are targeted by RBP10. All three proteins increase in procyclic forms but ZC3H20 appears earlier in differentiation [181]. RNAi targeting both ZC3H20 and ZC3H21 inhibits procyclic form growth [267] and ZC3H20 binds to, and stabilizes, mRNAs encoding some procyclic-specific membrane proteins. The role of ZC3H22 is unknown.

After differentiation from the bloodstream form, *T. brucei* initially grows as 'early' procyclic forms which express GPEET and EP procyclins. Later, GPEET mRNA is destabilized, and non-templated oligo-U tails were detected [268]. In opisthokonts, addition of oligo-U tails to mRNAs is a trigger for their 3′→5′ degradation by the enzyme Dis3L2 [269], and the U tails on GPEET are the first indication that this pathway exists in trypanosomes. Trypanosomes have a Dis3L2 homologue but its function is unknown and the terminal uridylyl transferase responsible for cytosolic U addition has not been identified (electronic supplementary material, Note 49).

## 5.8. Bloodstream-form-specific mRNAs

So far, no common sequence motifs or binding proteins have been found for mRNAs that are specific for the bloodstream form. A 16mer sequence in the *VSG* 3′-UTR is implicated in its stability in bloodstream forms [66]; in procyclic forms, the 3′-UTR gives a relatively normal half-life of about 25 min [270]. Selection for mutations that suppressed the function of the *GPIPLC* 3′-UTR in procyclic forms resulted in 3′ deletions, but the smallest regulatory 3′-UTR was 800 nt [271]. Results for the *PGKC* 3′-UTR implicated the 3′-most 360 nt in the suppression of expression in procyclic forms [272]. Oddly, this segment contains two copies of the RBP10 binding motif, despite the very high stability of *PGKC* mRNA in bloodstream forms.

The *TCTP* genes provide an example of competition between regulatory sequences. The *TCTP1* mRNA includes an RBP10 binding motif, and has very low expression in bloodstream forms, whereas the *TCTP2* mRNA shows opposite regulation. The first 160 nt of the *TCTP2* 3′-UTR are needed for optimal bloodstream-form expression; and when they are inserted upstream of the *TCTP1* 3′-UTR, developmental regulation of the latter is lost, suggesting that these 160 nt can prevent the effects of the RBP10 binding site [273].

## 5.9. From procyclic to metacyclic forms

RBP6 has a single RRM domain and its expression is highest in salivary gland *T. brucei* [274,275]. Inducible expression of RBP6 in procyclic forms triggered their differentiation to

epimastigotes then infectious metacyclic trypomastigotes [276]. Expression of a point mutant of RBP6 resulted in a 'jump' to infectious metacyclic forms without the appearance of epimastigotes [277]—which is also what happens after expression of RBP10 in procyclic forms. The direct RNA targets of RBP6 *in vivo* are not known, although it is tempting to speculate that it may stimulate RBP10 expression (electronic supplementary material, Note 50). Transcriptome and proteome results of the different stages [255,274,275,278] mainly reflect the transition from the procyclic to the bloodstream form, and growth arrest in metacyclic forms, but parasites with induced RBP6 also showed increases in mRNAs encoding RNA-binding proteins that might be specific to epimastigotes or metacyclic forms (electronic supplementary material, Note 51). One, RBP7, was also found to be needed for stumpy formation [279], so it might be implicated in developmental growth arrest.

DRBD13 is a repressor with several hundred identified bound mRNAs [280,281]. Both RNAi and over-expression killed procyclic forms, indicating that DRBD13 protein dosage is important [281]. The data were partially consistent with DRBD13 being a suppressor of RBP6 expression, but effects on the transcriptome after *DRBD13* RNAi were relatively minor [280,281] (electronic supplementary material, Note 52).

## 5.10. Abundant mRNA-binding proteins

Quantitative mass spectrometry results suggest that some RNA-binding proteins have abundances similar to those of the PABPs [181]. The ALBA proteins are top of the list, and were estimated by quantitative immunoblotting to be present at 10–20 000 molecules per procyclic trypanosome [282]. From the mass spectrometry results, UBP1 and UBP2, RBP42 and DRBD3/PTB1 are only slightly less abundant. To place this in context, a procyclic cell has about 44 000 mRNAs [45], and in *Leishmania* promastigotes the three PABPs are each present at around $1-2 \times 10^5$ molecules/cell [153]. V5-tagged UBP2 was estimated at over a million molecules per cell [283], but this seems anomalous—perhaps the protein was stabilized by the tag. Abundant RNA-binding proteins may be general RNA chaperones, although they could also have sequence-specific functions. The four cytoplasmic ALBA domain proteins mutually interact, and tandem affinity purification yielded PABP2, EIF4E4 and EIF4G3 [282], consistent with their being associated with translating mRNA. Pull-down of both PABPs reciprocally confirmed association with ALBA proteins [155].

The small RNA-binding proteins UBP1 and UBP2 prefer U-rich sequences [284], can homo- and hetero-dimerize [285] and show RNA-binding-dependent shuttling from the nucleus to the cytosol [286]. Only a few possible specific target mRNAs were detected [287–289] (electronic supplementary material, Note 53).

RBP42 and DRBD3/PTB1 are the only mRNA-binding proteins for which *in vivo* binding sites have been mapped in detail by identification of cross-linked nucleotides. RBP42 is essential in all tested life cycle stages, is found in the mRNP proteome, and stimulates when tethered. Binding sites in procyclic forms were particularly common in the coding regions of mRNAs of energy metabolism [290]. No consensus binding motif was reported. DRBD3 is also an activator of expression, and consistent with this, the 265 bound mRNAs are relatively

stable. DRBD3 binds predominantly in 3′-UTRs [143,290]. Microarrays detected changes in only eight of the 265 bound mRNAs after DRBD3 depletion [142,261], but the results did suggest that DRBD3 is important to maintain the stability of some bound mRNAs in procyclic forms [261] (electronic supplementary material, Note 54).

From mass spectrometry results, additional constitutively expressed abundant RNA-binding proteins are ZC3H41, DRBD2 and DRBD18, all of which are mRNA bound (electronic supplementary material, table S1). DRBD18 is described later.

## 5.11. Stress responses and ZC3H11

When eukaryotic cells are starved, over-heated or subjected to some chemical stresses, mRNAs and their associated proteins aggregate to form stress granules, which are thought to protect the mRNAs and enable renewed translation once normal conditions are restored [291,292] (figure 2 (9); electronic supplementary material, Note 55). Some effects of translation inhibition on mRNA decay may be caused by stress responses (electronic supplementary material, Note 56).

Most work on heat shock has been done with procyclic-form trypanosomes, which have a standard growth temperature of 27°C. After a severe heat shock (41°C), trypanosomes shut down transcription initiation [293]. In *T. cruzi*, this is accompanied (or perhaps caused) by RNA polymerase II large subunit dephosphorylation [294]. Translation of most *T. brucei* mRNAs is also suppressed, by an unknown mechanism, with either transfer to stress granules or degradation [295]. Effects are similar, but less extreme, at 39°C or the more physiological temperature of 37°C [296]. However, a subset of mRNAs, including those encoding all components of the protein refolding machinery, shows less granule association, and continues translation [296]. Initially, continued synthesis of these mRNAs is ensured by the fact that the genes are towards the ends of transcription units, so mRNAs can be made by elongating polymerases [297] (electronic supplementary material, Note 57). Meanwhile, maintenance of mRNA stability and translation is ensured by sequences in their 3′-UTRs.

The *HSP70* 3′-UTR is necessary and sufficient for the continued presence [298], stability and translation [295] of the mRNA after heat shock (electronic supplementary material, Note 57), and this requires a (UAU)n repeat which is bound by ZC3H11 and also found in other heat-shock protein transcripts. RNAi targeting *ZC3H11* caused death of bloodstream forms, while procyclic forms became highly susceptible to heat shock [299]. In heat-shocked procyclic forms, mRNAs that bind to ZC3H11 are preferentially retained on polysomes and excluded from heat-shock granules [296]. In both forms, ZC3H11 recruits an activating complex consisting of MKT1, PBP1, LSM12 and XAC1 [300]; PBP1 can recruit both PABPs. MKT1 complexes are probably implicated in the stabilization of many mRNAs, since MKT1 shows a two-hybrid interaction and/or *in vivo* association with 21 additional RNA-binding proteins, all but three of which activated when tethered (electronic supplementary material, table S1, sheet 1). Twelve of the interactors contain the motif H(E/D/N/Q)PY, which is necessary and sufficient for the interaction with MKT1, and several of the others have polyglutamine repeats (electronic supplementary material, table S1, sheet 1). MKT1 is essential in bloodstream

royalsocietypublishing.org/journal/rsob Open Biol. 9: 190072

royalsocietypublishing.org/journal/rsob   *Open Biol.* **9**: 190072

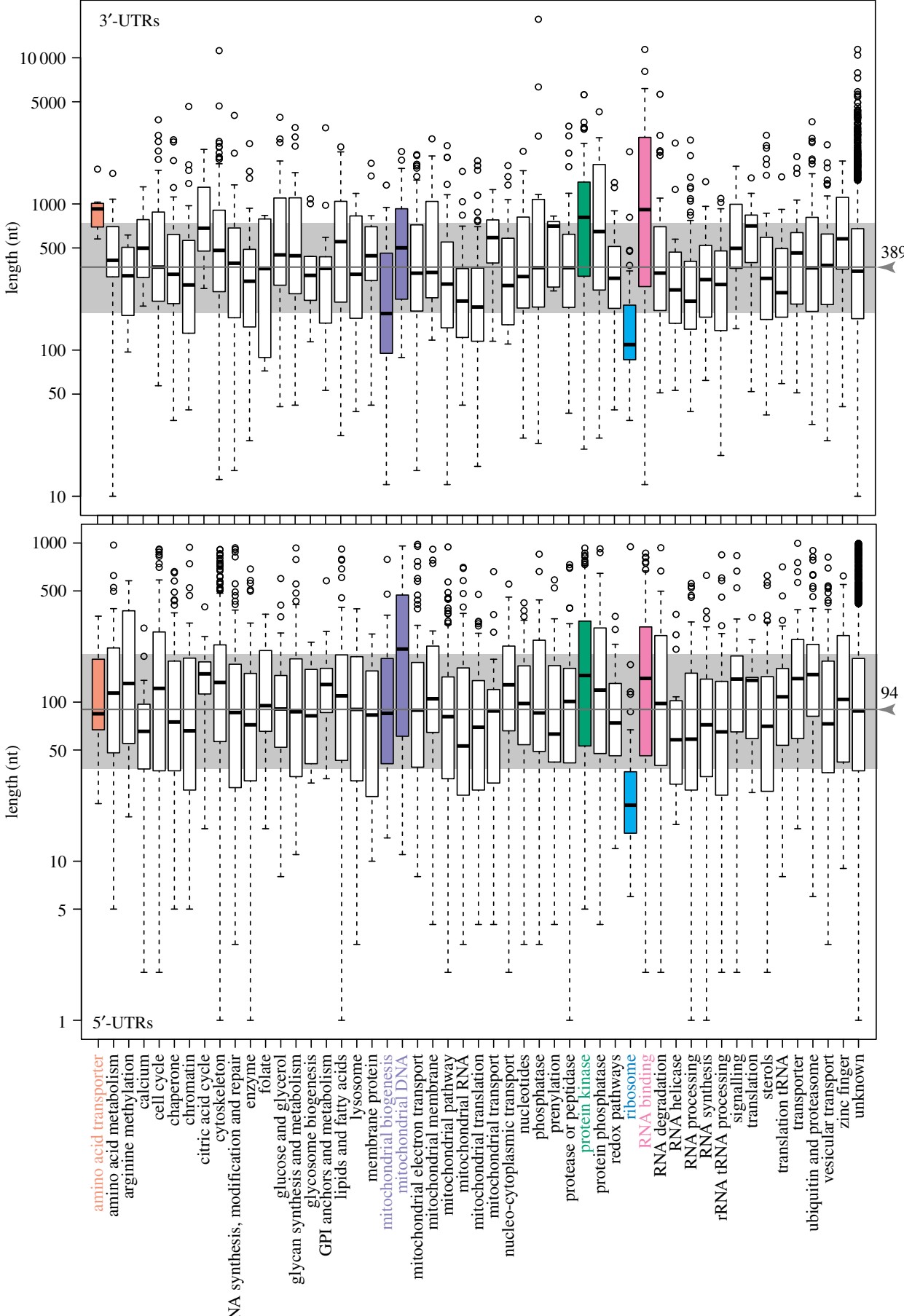

**Figure 6.** Lengths of 3′- and 5′-UTRs for mRNAs encoding proteins belonging to different functional classes. Protein classes were assigned manually, according to genome annotations and publications [171]. UTR lengths were downloaded from TritrypDB and are shown on a log scale (electronic supplementary material, Note 40). The boxes indicate the 25th and 75th percentiles, and the bar within the box is the median (label is in nt). Whiskers extend to the most extreme measurement that is within 1.5× the inter-quartile range, and spots are outliers. The dark grey line indicates the overall median, and the pale grey area is between the overall 25th and 75th quartiles. Colours indicate classes in which the medians for the 5′- and/or 3′-UTRs are outside the overall 25th–75th percentile range.

forms; roles in procyclic forms are unproven (electronic supplementary material, Note 58).

Trypanosome starvation granules and heat-shock granules contain much the same set of proteins as opisthokont stress granules [295,301–303]. Interestingly, only PABP2, but not PABP1, accumulates in stress granules [154]. SCD6, an evolutionarily conserved granule component, functions as a translation repressor in trypanosomes [304]; upon over-expression, it aggregates and promotes granule formation [305] (electronic supplementary material, Note 59). Curiously, starvation causes ALPH1 and XRNA to move to a specialized stress granule at the posterior end of the cell [303]. The significance of this is unknown.

## 5.12. The odd behaviour of mRNAs encoding ribosomal proteins

When cells are stressed, ribosomal protein synthesis is not required. It is, therefore, curious that the mRNAs encoding *T. brucei* ribosomal proteins are excluded from both starvation and heat-shock granules [296,303]. Ribosomal protein mRNAs are very stable but have abnormally low ribosome occupancy (electronic supplementary material, Note 60). This might be because they have very good codon optimality scores [196], which are expected to result in rapid elongation [306]. These mRNAs are also notable for their short 5′- and 3′-UTRs (figure 6). So far, the only RNA-binding protein known to be associated with these mRNAs is DRBD3 [143] (electronic supplementary material, Note 61).

## 5.13. Regulating the regulators

As RNA-binding proteins are vital for gene expression control, it is essential that they too are tightly regulated. More than 30 *T. brucei* RNA-binding proteins show at least twofold differences in synthesis between bloodstream and procyclic forms [126,193], with contributions from both mRNA level and translation efficiency. Even more are regulated at the protein level [181], where proteolysis can play a role. In addition, more than half are phosphorylated [24], and 11 have methylated arginine residues [307] (electronic supplementary material, table S1). At least one is SUMOylated (as are some exosome subunits) [308], and several others are acetylated [309] (electronic supplementary material, table S1).

Intriguingly, the mRNAs encoding many RNA-binding proteins have unusually long 3′-UTRs—often several kb long—and this is also true for protein kinases (figure 6). There could be two reasons for this. One is the inverse correlation between abundance and mRNA length: the long 3′-UTR may help to ensure low mRNA abundance. The other possibility is that because the long 3′-UTRs can bind many regulatory proteins, control can be more stringent or flexible than for shorter 3′-UTRs. The *ZC3H20*, *ZC3H21* and *ZC3H22*

mRNAs are a clear example, with 3′-UTRs longer than 4 kb. *ZC3H20* mRNA has two RBP10 binding sites, and ZC3H20 becomes detectable during stumpy formation [181]; but the other two mRNAs have five and nine binding sites, respectively [252], and the proteins are detected only later in differentiation, when RBP10 has disappeared [181]. The presence of so many binding sites for the same RNA-binding protein is suggestive of a fail-safe mechanism which ensures that the mRNA is destroyed even if polyadenylation occurs upstream of the dominant sites.

ZC3H11 expression increases rapidly upon heat shock—but this has to happen with very little mRNA synthesis. Instead, translation of existing *ZC3H11* mRNA is strongly upregulated, governed by a 70 nt region in the 3′-UTR. ZC3H11 protein stability also increases, probably as a consequence of dephosphorylation [310].

DRBD18 is a substrate of the arginine methyltransferase PRMT1 [311] (electronic supplementary material, Note 62). Depletion of DRBD18 in procyclic forms resulted mainly in mRNA increases which were suggestive of at least partial procyclic–epimastogote–metacyclic differentiation. Whether this effect was direct or indirect is not clear. Further results indicated that methylation promotes mRNA stabilization by DRBD18, as well as changes its RNA-binding specificity and protein–protein interactions [311].

# 6. Future challenges

Numerous gaps in our knowledge have been mentioned in this review. We have no detailed information concerning regulation of mRNA processing, and do not know how the mature mRNA 5′-end is recognized for export from the nucleus. Very few kinetoplastid RNA-binding proteins have been quantitatively analysed using modern high-throughput methods; we do not know how they interact, or compete with each other, or, in most cases, how their activities are regulated by post-translational modification. Sequence-specific recruitment of the mRNA decay apparatus is a mystery: no interactions with destabilizing RNA-binding proteins have been found. We also do not know why there are five different translation-promoting cap-binding complexes. The biggest challenge, however, is disentangling the numerous interactions that can occur on a single mRNA.

Even in organisms that have transcription control many of the same questions remain to be answered, and it is quite possible that trypanosomatid solutions to these problems are conserved—or have evolved separately—in other supergroups.

Data accessibility. This article has no additional data.

Competing interests. I declare I have no competing interests.

Funding. I received no funding for this study.

Acknowledgements. Work in my laboratory is funded by the Deutsche Forschungsgemeinschaft and by the State of Baden-Württemberg.

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
