## [Reviewer comments · Open Biology]

Review History

RSOB-19-0072.R0 (Original submission)

Review form: Reviewer 1 (Arthur Gunzl)

Recommendation

Accept as is

Are each of the following suitable for general readers?

- a) **Title**
Yes
- b) **Summary**
Yes
- c) **Introduction**
Yes

Is the length of the paper justified?

Yes

Should the paper be seen by a specialist statistical reviewer?

No

Is it clear how to make all supporting data available?

Yes

Is the supplementary material necessary; and if so is it adequate and clear?

Yes

Do you have any ethical concerns with this paper?

No

Comments to the Author

Please see Appendix A

Review form: Reviewer 2

Recommendation

Major revision is needed (please make suggestions in comments)

Are each of the following suitable for general readers?

- a) **Title**
No
- b) **Summary**
Yes
- c) **Introduction**
No

Is the length of the paper justified?

No

Should the paper be seen by a specialist statistical reviewer?

No

Is it clear how to make all supporting data available?

Not Applicable

Is the supplementary material necessary; and if so is it adequate and clear?

No

Do you have any ethical concerns with this paper?

No

Comments to the Author

I was surprised by the title of this manuscript “How to live without transcription factors: lessons from the trypanosomatids” for two reasons. First, I expected it to centre on transcription, but this is only a very minor part of the whole. In many respects, it is an update of a similar review on gene expression by this author in 2016. Second, there is evidence - some of it very recent, some of it dating back ten years - that these organisms do have transcription factors. Some of the basal transcription factors were not identified previously as they are highly divergent (Srinistava et al., 2018), others are recognisable orthologues of transcription elongation factors in other eukaryotes, and a third set of factors involved in transcription are unique. One way to solve this would be to change the title to something along the lines of “How to live with polycistronic transcription....”. Alternatively, the author could write a more focused review outlining how concepts of transcription in trypanosomes are evolving – this would be something new.

Major points:

The section on transcription would benefit enormously from being expanded to give a more comprehensive account of findings in recent publications. Since the review is already rather long, this could be balanced by condensing more speculative sections on RNA-binding proteins of unknown function.

1. Transcription of protein-coding genes by RNA polymerase I (in particular the VSG expression site in bloodstream forms) is partly tucked into the section on genome organisation and transcription. The section on transcription by RNAPI is extremely brief (only half a page). Specific factors required for transcription of bloodstream and metacyclic VSGs are not mentioned. The whole issue of monoallelic exclusion is summarised in very general terms in just two sentences. I disagree with the author that this is beyond the scope of this review. It should be a central part of how trypanosomes can regulate expression with or without conventional transcription factors.
2. RNA polymerase II. This section mentions the basal transcription factors recently identified by Srinistava et al. and some transcription elongation factors, but in a rather cursory way. It should at least be mentioned that they were identified because of their requirement for transcription of the spliced leader precursor, which is also by RNAPII. It might be worth rephrasing a negative (it is likely that they are used for polycistronic transcription, but this has not been tested) with a positive (this is something that can now be tested fairly easily with ChIP-seq). It has recently been shown that some RHS proteins bind both to spliced leader and polycistronic RNAPII transcription units (see point 5).
3. The role of histone variants as markers of transcription start sites and termination sites (Siegel et al., 2009) also deserves more thorough treatment as the way in which trypanosomatids, in contrast to many other eukaryotes, define their transcription units.
4. The only reference to nascent RNA synthesis comes from an analysis of a region of *Leishmania* chromosome 3 (Martinez-Calvillo et al., 2004) and omits a recent reference to the establishment of GRO-Seq in trypanosomes (Florini et al., 2019).
5. The RHS proteins are dealt with extremely briefly and some recent data is omitted (Florini et al., 2019). There are 7 RHS sub-families according to TritrypDB. The last one was not identified in the original paper by Bringaud et al. (2002), most probably because it is barely expressed in bloodstream and procyclic forms. RHS6 is not the only chromatin-associated RHS – RHS2 and RHS4 are as well. Their co-localisation with RNAP II, analysed by ChIP-Seq, and their global requirement for nascent RNA synthesis suggests that they are likely to be transcription elongation factors. Since RHS only occur in the African and South American trypanosomes, this suggests that these organisms have co-opted their own unique transcription factors. Might

Leishmania, Crithidia, etc have done something similar? A review seems a good place to raise such questions.

6. Unpublished results from the author's lab are cited on 9 occasions. A couple of paragraphs are composed almost entirely of unpublished data. What is the journal policy on this?

Minor points:

7. page 2. *T. cruzi* epimastigotes are not the only forms in reduviid bugs. Metacyclics in the hindgut are excreted and infect a new host.

8. page 8, line 1. Trans splicing sites are usually preceded by a polypyrimidine tract. Are there any exceptions to this?

9. The section on alternative splicing is quite hard to follow, as is the figure. This section misses references for UTRs affecting RNA turnover. A separate small section on the role of UTRs comes on page 15. Wouldn't it make sense to fuse these?

10. TRRM1 was shown to be essential in refs 129 and 131, before ref 130. The author focuses on its RNA-binding capacity, but this ignores that it associates with PTB2 and RHS, which also ties it to transcription. It also seems to be a chromatin remodeller and one of the major effects of depletion was an increase in heterochromatin.

11. page 11. NMD3, which is also involved in mRNA export in trypanosomes, is not mentioned.

12. page 14. UFP1 seems to have a lot of other functions these days (see, for example, Kim and Maquat, 2018).

13. page 21. Abundant RNA-binding proteins – can the author put numbers on these? Some have been measured in the past.

14. page 22. Stress responses. Chromatin compaction is another response to heat shock (ref 131). This could explain reduced transcription.

15. page 25. Since gene expression control is mediated by RBPs.... This is too narrow. By now there is ample evidence that epigenetic and transcription factors are also important.

16. It is not clear why standard terms are often put in inverted commas.

17. There are a few typos in the manuscript, e.g. urudyliyl, becuase...

Figures

Figure 1. It is not clear why "metacyclic" is written in grey, and all other stages in black.

Figure 2. The author is entitled to reproduce a figure that she has published before – no problems with that. The issues here are that it is extremely complicated and does not reflect recent findings (e.g. co-transcriptional nuclear export of mRNA, Goos et al., 2018, participation of RHS in transcription). On the same page as the figure (p.3) processing is referred to as post-transcriptional.

Wouldn't it be more appropriate to have a figure summarising what is known about mRNA transcription by RNAPI and II?

Figure 3 is also very complicated for a general audience. The majority of splicing and processing events are of type B, not type C. One could leave it at that.

Figure 4. I could not work out which results were from *Leishmania* and which were from *T. brucei*. Maybe this could be colour-coded. At the same time, the activator/repressor colour code could be removed. These data come from tethering assays and it is not clear if they also apply under normal circumstances. Unpublished data from the author's lab is apparently included. It is not clear how extensive this is in this particular figure - it will also depend on the journal's policy (see point 6).

Figure 5. It would be helpful to use the same colours in this figure and figure 1. It is not clear where the expression data come from. Do they refer to RNA or protein? How much is published/unpublished? The different scales of blue and grey are not so obvious on a printout - increasing contrast or employing different colours would help.

Decision letter (RSOB-19-0072.R0)

26-Apr-2019

Dear Dr Clayton

We are pleased to inform you that your manuscript RSOB-19-0072 entitled "How to live without transcription factors: lessons from the trypanosomatids" has been accepted by the Editor for publication in *Open Biology*. The reviewer(s) have recommended publication, but also suggest some minor revisions to your manuscript. Therefore, we invite you to respond to the reviewer(s)' comments and revise your manuscript.

Please submit the revised version of your manuscript within 14 days. If you do not think you will be able to meet this date please let us know immediately and we can extend this deadline for you.

- 1) A text file of the manuscript (doc, txt, rtf or tex), including the references, tables (including captions) and figure captions. Please remove any tracked changes from the text before submission. PDF files are not an accepted format for the "Main Document".
- 2) A separate electronic file of each figure (tiff, EPS or print-quality PDF preferred). The format should be produced directly from original creation package, or original software format. Please note that PowerPoint files are not accepted.
- 3) Electronic supplementary material: this should be contained in a separate file from the main text and meet our ESM criteria (see <http://royalsocietypublishing.org/instructions-authors#question5>). All supplementary materials accompanying an accepted article will be treated as in their final form. They will be published alongside the paper on the journal website and posted on the online figshare repository. Files on figshare will be made available approximately one week before the accompanying article so that the supplementary material can be attributed a unique DOI.

Online supplementary material will also carry the title and description provided during submission, so please ensure these are accurate and informative. Note that the Royal Society will not edit or typeset supplementary material and it will be hosted as provided. Please ensure that the supplementary material includes the paper details (authors, title, journal name, article DOI). Your article DOI will be 10.1098/rsob.2016[last 4 digits of e.g. 10.1098/rsob.20160049].

- 4) A media summary: a short non-technical summary (up to 100 words) of the key findings/importance of your manuscript. Please try to write in simple English, avoid jargon, explain the importance of the topic, outline the main implications and describe why this topic is newsworthy.

Images

Data-Sharing

It is a condition of publication that data supporting your paper are made available. Data should be made available either in the electronic supplementary material or through an appropriate repository. Details of how to access data should be included in your paper. Please see <http://royalsocietypublishing.org/site/authors/policy.xhtml#question6> for more details.

Data accessibility section

Sincerely,

The Open Biology Team

<mailto:openbiology@royalsociety.org>

Editor's comment:

Hi Christine: You'll see from the referees' reports that one is more favorable than the other. I think you can deal with these comments as minor revisions but please take the comments into account and let me know how you have addressed them.

This referee also remarks on your inclusion of unpublished data. I have no problem with this...although we could take care of it by calling the article a perspective (or even "retrospective") rather than a review.

Please get in touch with me directly if you would like to discuss this further.

Thanks again

David

Reviewer(s)' Comments to Author:

Referee: 1

Comments to the Author(s)

Please see attached file

Referee: 2

Comments to the Author(s)

I was surprised by the title of this manuscript "How to live without transcription factors: lessons from the trypanosomatids" for two reasons. First, I expected it to centre on transcription, but this is only a very minor part of the whole. In many respects, it is an update of a similar review on gene expression by this author in 2016. Second, there is evidence - some of it very recent, some of it dating back ten years - that these organisms do have transcription factors. Some of the basal transcription factors were not identified previously as they are highly divergent (Srinistava et al., 2018), others are recognisable orthologues of transcription elongation factors in other eukaryotes, and a third set of factors involved in transcription are unique. One way to solve this would be to change the title to something along the lines of "How to live with polycistronic transcription....". Alternatively, the author could write a more focused review outlining how concepts of transcription in trypanosomes are evolving - this would be something new.

Major points:

The section on transcription would benefit enormously from being expanded to give a more comprehensive account of findings in recent publications. Since the review is already rather long, this could be balanced by condensing more speculative sections on RNA-binding proteins of unknown function.

1. Transcription of protein-coding genes by RNA polymerase I (in particular the VSG expression site in bloodstream forms) is partly tucked into the section on genome organisation and transcription. The section on transcription by RNAPI is extremely brief (only half a page). Specific factors required for transcription of bloodstream and metacyclic VSGs are not mentioned. The whole issue of monoallelic exclusion is summarised in very general terms in just two sentences. I disagree with the author that this is beyond the scope of this review. It should be a central part of how trypanosomes can regulate expression with or without conventional transcription factors.

2. RNA polymerase II. This section mentions the basal transcription factors recently identified by Srinistava et al. and some transcription elongation factors, but in a rather cursory way. It should at least be mentioned that they were identified because of their requirement for transcription of

the spliced leader precursor, which is also by RNAPII. It might be worth rephrasing a negative (it is likely that they are used for polycistronic transcription, but this has not been tested) with a positive (this is something that can now be tested fairly easily with CHIP-seq). It has recently been shown that some RHS proteins bind both to spliced leader and polycistronic RNAPII transcription units (see point 5).

3. The role of histone variants as markers of transcription start sites and termination sites (Siegel et al., 2009) also deserves more thorough treatment as the way in which trypanosomatids, in contrast to many other eukaryotes, define their transcription units.
4. The only reference to nascent RNA synthesis comes from an analysis of a region of *Leishmania* chromosome 3 (Martinez-Calvillo et al., 2004) and omits a recent reference to the establishment of GRO-Seq in trypanosomes (Florini et al., 2019).
5. The RHS proteins are dealt with extremely briefly and some recent data is omitted (Florini et al., 2019). There are 7 RHS sub-families according to TritypDB. The last one was not identified in the original paper by Bringaud et al. (2002), most probably because it is barely expressed in bloodstream and procyclic forms. RHS6 is not the only chromatin-associated RHS – RHS2 and RHS4 are as well. Their co-localisation with RNAP II, analysed by CHIP-Seq, and their global requirement for nascent RNA synthesis suggests that they are likely to be transcription elongation factors. Since RHS only occur in the African and South American trypanosomes, this suggests that these organisms have co-opted their own unique transcription factors. Might *Leishmania*, *Crithidia*, etc have done something similar? A review seems a good place to raise such questions.
6. Unpublished results from the author's lab are cited on 9 occasions. A couple of paragraphs are composed almost entirely of unpublished data. What is the journal policy on this?

Minor points:

7. page 2. *T. cruzi* epimastigotes are not the only forms in reduviid bugs. Metacyclics in the hindgut are excreted and infect a new host.
8. page 8, line 1. Trans splicing sites are usually preceded by a polypyrimidine tract. Are there any exceptions to this?
9. The section on alternative splicing is quite hard to follow, as is the figure. This section misses references for UTRs affecting RNA turnover. A separate small section on the role of UTRs comes on page 15. Wouldn't it make sense to fuse these?
10. TRRM1 was shown to be essential in refs 129 and 131, before ref 130. The author focuses on its RNA-binding capacity, but this ignores that it associates with PTB2 and RHS, which also ties it to transcription. It also seems to be a chromatin remodeller and one of the major effects of depletion was an increase in heterochromatin.
11. page 11. NMD3, which is also involved in mRNA export in trypanosomes, is not mentioned.
12. page 14. UFP1 seems to have a lot of other functions these days (see, for example, Kim and Maquat, 2018).
13. page 21. Abundant RNA-binding proteins – can the author put numbers on these? Some have been measured in the past.

14. page 22. Stress responses. Chromatin compaction is another response to heat shock (ref 131). This could explain reduced transcription.

15. page 25. Since gene expression control is mediated by RBPs.... This is too narrow. By now there is ample evidence that epigenetic and transcription factors are also important.

16. It is not clear why standard terms are often put in inverted commas.

17. There are a few typos in the manuscript, e.g. urudyliyl, becuae...

Figures

Figure 1. It is not clear why “metacyclic” is written in grey, and all other stages in black.

Figure 2. The author is entitled to reproduce a figure that she has published before – no problems with that. The issues here are that it is extremely complicated and does not reflect recent findings (e.g. co-transcriptional nuclear export of mRNA, Goos et al., 2018, participation of RHS in transcription). On the same page as the figure (p.3) processing is referred to as post-transcriptional.

Wouldn't it be more appropriate to have a figure summarising what is known about mRNA transcription by RNAPI and II?

Figure 3 is also very complicated for a general audience. The majority of splicing and processing events are of type B, not type C. One could leave it at that.

Figure 4. I could not work out which results were from *Leishmania* and which were from *T. brucei*. Maybe this could be colour-coded. At the same time, the activator/repressor colour code could be removed. These data come from tethering assays and it is not clear if they also apply under normal circumstances. Unpublished data from the author's lab is apparently included. It is not clear how extensive this is in this particular figure - it will also depend on the journal's policy (see point 6).

Figure 5. It would be helpful to use the same colours in this figure and figure 1.

It is not clear where the expression data come from. Do they refer to RNA or protein? How much is published/unpublished?

The different scales of blue and grey are not so obvious on a printout – increasing contrast or employing different colours would help.

Author's Response to Decision Letter for (RSOB-19-0072.R0)

See Appendix B.

Decision letter (RSOB-19-0072.R1)

10-May-2019

Dear Dr Clayton

We are pleased to inform you that your manuscript entitled "Regulation of gene expression in trypanosomatids: living with polycistronic transcription" has been accepted by the Editor for publication in Open Biology.

Sincerely,

The Open Biology Team
mailto: openbiology@royalsociety.org

Appendix A

The review article by Christine Clayton covers a tremendous amount of gene expression work in trypanosomes and related organisms. It is a valuable and up-to-date review that includes transcriptional aspects and focuses on RNA metabolism, providing an astonishing amount of detail for gene expression steps and pathways which often deviate significantly from those of the model organisms. The review is well written and, undoubtedly, will be a great resource for researchers in and outside the field of Molecular Parasitology. This reviewer has one major criticism and a few suggestions for Dr. Clayton to improve the article.

Major Concern

- 1 This reviewer does not support the title because it is misleading. Trypanosomes possess all basal transcription initiation factors for RNA pol II, TFIIB for RNA pol III, and an essential factor for RNA pol I. So far, ~40 proteins have been identified as part of essential transcription initiation factors. There are likely additional transcription factors such as TFIIIA and TFIIIC that have not been identified yet because they are extremely divergent from their eukaryotic counterparts. Dr. Clayton has published this misconception before and there are numerous articles which have reiterated this fact without reflection. What she really means though is that trypanosomes lack DNA-binding transcription activators and repressors which account for much of gene expression regulation in higher eukaryotes. Despite bringing a resonating quality to the title, the generic expression “transcription factors” is not ideal.

Suggestions

- 2 Introduction, 2nd paragraph. The way the genera names are used here is awkward. How about “...they include the plant parasites of the genus *Phytomonas*, and *Leishmania* and *Trypanosoma* species which cause diseases in vertebrates.”
- 3 Page 5, first paragraph. In both cited references 16 & 41, the complex studied also contained the unequivocal homolog of the small TFIIA subunit. This complex comprises SNAPc, TBP and TFIIA with one or two unique components, depending on whether you consider the third SNAP protein of SNAPc to be the homolog of human SNAP190 or not. TFIIA should be included in the text though.

Although this may be too much detail: all the other general transcription factors were shown to be essential for SL RNA gene transcription in vitro as well, including TFIIH. As shown in other systems, TFIIH has helicase activity that, in addition to the described kinase function, is required to separate the DNA strands around the transcription initiation site.

Naming of TRF4/TBP: From the beginning, the TBP of trypanosomes was named TRF4 (Tschudi/Ullu fraction and original naming) or [t]TBP (George Cross fraction). However, this reviewer has recently heard talks about the different trypanosome protein “TRF4 polyA polymerase”. Since all functional data so far are in line with TRF4 being the TBP homolog in trypanosomes, I suggest to use the name TBP and say “originally termed TRF4”. In the long run, this will be the better choice.

- 4 Note 12: The author should include the branch point mapping study by the Bindereif group. Lücke *et al.*, 205, MBP 142:248-251.
- 5 page 8, last paragraph. It is correct that there is no data which would suggest direct interaction of the spliceosome with RNA pol II. However, there is evidence for cross-talk between trans splicing and RNA pol II because depletion of spliceosomal components, f. ex. of PRP19 subunits (Ambrosio *et al.*, 2015, Mol Microbiol), resulted in the loss of phosphorylation of RPB1, the functional consequence of which is still unclear.

- 6 page 10 last paragraph: The sentence “U2AF35, U2AF65, and SF1 depletion also affected both splicing (at a global level) and stability [138].” seems to be misplaced because the paragraph deals with HNRNPH/F.
- 7 Figure 2 step 7 & page 11 last paragraph. The figure indicates that it is known that the exon junction complex is deposited on SL *trans*-spliced mRNA. However, to the best of my knowledge, this has not been shown and it is also not evident from the description in the paragraph and the corresponding notes (maybe this complex is only important for cis splicing). If not shown yet, I think the wording, especially in Fig. 2, needs to be adapted such that it becomes clear that this is a hypothesis – otherwise readers will take it for granted.
- 8 The supplemental tables are not well introduced in the main text. There needs to be a better explanation of what these lists are and how they were derived. How comprehensive are they? For example, looking closer at the “4. Helicase” list, the DBP1/HEL64 annotation of Tb927.10.6630 seems odd because, in TriTrypDB, this helicase is annotated as DBP2A and, accordingly, blasting its sequence against the human/yeast data bases revealed, as closest matches, human DDX5 and its yeast homolog DBP2. In addition, the paralog of DBP2A, DBP2B (Tb927.8.1510), which is clearly a DEAD box RNA helicase, is not listed at all.

Appendix B

I thank the reviewers for their input, especially regarding the subjects where my knowledge is rather weaker. Their help has been invaluable in preventing omissions.

1. Title. The reviewers are absolutely right, this is really not an appropriate title and I have changed it according to one of the suggestions

PDF review

2 Introduction, 2nd paragraph. The way the genera names are used here is awkward. How about "...they include the plant parasites of the genus *Phytomonas*, and *Leishmania* and *Trypanosoma* species which cause diseases in vertebrates."

Changed

3 Page 5, first paragraph. In both cited references 16 & 41, the complex studied also contained the unequivocal homolog of the small TFIIA subunit. This complex comprises SNAPc, TBP and TFIIA with one or two unique components, depending on whether you consider the third SNAP protein of SNAPc to be the homolog of human SNAP190 or not. TFIIA should be included in the text though.

Although this may be too much detail: all the other general transcription factors were shown to be essential for SL RNA gene transcription in vitro as well, including TFIIH. As shown in other systems, TFIIH has helicase activity that, in addition to the described kinase function, is required to separate the DNA strands around the transcription initiation site.

Changed

Naming of TRF4/TBP: From the beginning, the TBP of trypanosomes was named TRF4 (Tschudi/Ullu fraction and original naming) or [t]TBP (George Cross fraction). However, this reviewer has recently heard talks about the different trypanosome protein "TRF4 polyA polymerase". Since all functional data so far are in line with TRF4 being the TBP homolog in trypanosomes, I suggest to use the name TBP and say "originally termed TRF4". In the long run, this will be the better choice.

Changed

4 Note 12: The author should include the branch point mapping study by the Bindereif group. Lücke et al., 205, MBP 142:248-251.

Done, also in the main text.

5 page 8, last paragraph. It is correct that there is no data which would suggest direct interaction of the spliceosome with RNA pol II. However, there is evidence for cross-talk between trans splicing and RNA pol II because depletion of spliceosomal components, f. ex. of PRP19 subunits (Ambrosio et al., 2015, Mol Microbiol), resulted in the loss of phosphorylation of RPB1, the functional consequence of which is still unclear.

There is a chicken-and-egg problem here which will be difficult to resolve. Depletion of either CRK9 or PRP19 results in decreased cap methylation. (I couldn't find the evidence that depletion of PRP19 results in decreased pol II phosphorylation.) But its depletion of also inhibits splicing (resulting in an increase in "unused" SLRNA) and impairs growth. There's a similar problem with CRK9 depletion. The text has been altered to include this.

6 page 10 last paragraph: The sentence "U2AF35, U2AF65, and SF1 depletion also affected both splicing (at a global level) and stability [138]." seems to be misplaced because the paragraph deals with HNRNPH/F.

I moved this to an earlier position.

7 Figure 2 step 7 & page 11 last paragraph. The figure indicates that it is known that the exon junction complex is deposited on SL trans-spliced mRNA. However, to the best of my knowledge, this has not been shown and it is also not evident from the description in the paragraph and the corresponding notes (maybe this complex is only important for cis splicing). If not shown yet, I think the wording, especially in Fig. 2, needs to be adapted such that it becomes clear that this is a hypothesis – otherwise readers will take it for granted.

True. I've added a caveat to the Figure legend and the text.

8 The supplemental tables are not well introduced in the main text. There needs to be a better

explanation of what these lists are and how they were derived. How comprehensive are they? For example, looking closer at the "4. Helicase" list, the DBP1/HEL64 annotation of Tb927.10.6630 seems odd because, in TriTrypDB, this helicase is annotated as DBP2A and, accordingly, blasting its sequence against the human/yeast data bases revealed, as closest matches, human DDX5 and its yeast homolog DBP2. In addition, the paralog of DBP2A, DBP2B (Tb927.8.1510), which is clearly a DEAD box RNA helicase, is not listed at all.

This sheet now includes only the DEAD/H-box helicases that are in the cytoplasm. There are lots in the nucleus (often, the nucleolus) and only a few have known functions. I think some of the names stem from when the whole genome wasn't available. Someone (not me) would need to do some proper phylogeny on all these proteins in order to find out what they are doing.

The Legend now describes criteria used to choose the proteins that are in the different Tables, and both the Notes and the Table are introduced at the beginning of the review.

REVIEWER 2

Major points:

The section on transcription would benefit enormously from being expanded to give a more comprehensive account of findings in recent publications. Since the review is already rather long, this could be balanced by condensing more speculative sections on RNA-binding proteins of unknown function.

I agree that this would be interesting but I don't think that I am the ideal person. Maybe the reviewer might wish to do this once she has more analysis of the GRO-Seq results? Other possible authors are Arthur Günzl and Nicolai Siegel.

The review description at the beginning makes it clear that transcription is not covered in as much detail as some other topics.

1. Transcription of protein-coding genes by RNA polymerase I (in particular the VSG expression site in bloodstream forms) is partly tucked into the section on genome organisation and transcription. The section on transcription by RNAPI is extremely brief (only half a page). Specific factors required for transcription of bloodstream and metacyclic VSGs are not mentioned. The whole issue of monoallelic exclusion is summarised in very general terms in just two sentences. I disagree with the author that this is beyond the scope of this review. It should be a central part of how trypanosomes can regulate expression with or without conventional transcription factors.

I think that reviewing all this literature would add several pages (and another month of work). The field is moving really quickly (lots of new factors in a recent meeting abstract book) and someone else could be invited to review it. How about Gloria Rudenko? I think she would give a really balanced assessment of results from different labs.

2. RNA polymerase II. This section mentions the basal transcription factors recently identified by Srinistava et al. and some transcription elongation factors, but in a rather cursory way. It should at least be mentioned that they were identified because of their requirement for transcription of the spliced leader precursor, which is also by RNAPII. It might be worth rephrasing a negative (it is likely that they are used for polycistronic transcription, but this has not been tested) with a positive (this is something that can now be tested fairly easily with ChIP-seq).

Comment added.

It has recently been shown that some RHS proteins bind both to spliced leader and polycistronic RNAPII transcription units (see point 5).

The roles of these proteins are mentioned but mechanisms are still very unclear.

3. The role of histone variants as markers of transcription start sites and termination sites (Siegel et al., 2009) also deserves more thorough treatment as the way in which trypanosomatids, in contrast to many

other eukaryotes, define their transcription units.

The relevant variants are all listed - it's not clear to me what else there is to say. The "readers" haven't been investigated yet, so far as I know.

4. The only reference to nascent RNA synthesis comes from an analysis of a region of Leishmania chromosome 3 (Martinez-Calvillo et al., 2004) and omits a recent reference to the establishment of GRO-Seq in trypanosomes (Florini et al., 2019).

The paper in which GRO-Seq is described gives no details about the distribution of reads over chromosomes, mentioning only that the method is reproducible and that incorporation is inhibited by depletion of the RHS proteins. The reads are available but I assume that the authors intend to publish another, more detailed study. For me to do this in the context of the review would not be appropriate. In the meantime I have mentioned that the method exists. This is covered more fully in note 38/

5. The RHS proteins are dealt with extremely briefly and some recent data is omitted (Florini et al., 2019). There are 7 RHS sub-families according to TritrypDB. The last one was not identified in the original paper by Bringaud et al. (2002), most probably because it is barely expressed in bloodstream and procyclic forms. RHS6 is not the only chromatin-associated RHS – RHS2 and RHS4 are as well. Their co-localisation with RNAP II, analysed by CHIP-Seq, and their global requirement for nascent RNA synthesis suggests that they are likely to be transcription elongation factors. Since RHS only occur in the African and South American trypanosomes, this suggests that these organisms have co-opted their own unique transcription factors. Might Leishmania, Crithidia, etc have done something similar? A review seems a good place to raise such questions.

A little more detail has been added about these proteins, but this is just one of 290 cited papers (not including the Notes). The mechanisms by which the RHS proteins act are not known so adding any more would be disproportionate.

6. Unpublished results from the author's lab are cited on 9 occasions. A couple of paragraphs are composed almost entirely of unpublished data. What is the journal policy on this?

In two cases (amounts of the translation factors) I realised I could replace this with a citation of published results. In other cases I have removed the comments.

Minor points:

7. page 2. T. cruzi epimastigotes are not the only forms in reduviid bugs. Metacyclics in the hindgut are excreted and infect a new host.

Corrected.

8. page 8, line 1. Trans splicing sites are usually preceded by a polypyrimidine tract. Are there any exceptions to this? Yes, quite a lot - depending on how you define a polypyrimidine tract, of course.

9. The section on alternative splicing is quite hard to follow, as is the figure. This section misses references for UTRs affecting RNA turnover. A separate small section on the role of UTRs comes on page 15. Wouldn't it make sense to fuse these?

I don't think they should be fused because then degradation/translation and splicing get mixed up. The sentence has been re-worded.

10. TRRM1 was shown to be essential in refs 129 and 131, before ref 130. The author focuses on its RNA-binding capacity, but this ignores that it associates with PTB2 and RHS, which also ties it to transcription. It also seems to be a chromatin remodeller and one of the major effects of depletion was an

increase in heterochromatin.

Ref 129 (Manger & Boothroyd, 1998) doesn't examine whether TRRM1 is essential or not, but I've added ref 131. PTB2 has also been postulated to be a splicing factor. The increase in heterochromatin could be secondary since the results were obtained at a point when the cells had stopped growing.

11. page 11. NMD3, which is also involved in mRNA export in trypanosomes, is not mentioned.

I am reluctant to cite these results because all analyses were done after 3 days' RNAi induction, whereas growth inhibition was apparent after only 24h and was strong after 2 days. Moreover it would mean having to explain what PAG genes are. As a compromise, I have added NMD3 to the Supplementary Table 1 sheet on mRNA export.

12. page 14. UFP1 seems to have a lot of other functions these days (see, for example, Kim and Maquat, 2018).

Thank you for pointing out the existence of this review, which I had missed and have now cited. Of course UPF1 in tryps may be doing something different, but as yet, we don't know what.

13. page 21. Abundant RNA-binding proteins – can the author put numbers on these? Some have been measured in the past.

I found the result for ALBA proteins. You can get a rough idea from quantitative label-free proteomics (Dejung et al, now cited) although this will depend heavily on the number of detectable peptides. The published results have Lfq values, but I did some calculations based on total CDS length and the results are similar to those from our own unpublished iBAQ values.

14. page 22. Stress responses. Chromatin compaction is another response to heat shock (ref 131). This could explain reduced transcription.

Supplementary Figure S5 in Ref 131 shows that heat shock resulted in 1.5-3-fold increased DNA pull-down using antibody to histone H3 - with an anomalously high result for the HSP70 genes that is so far unexplained. This is now discussed in the Notes. (Incidentally, the legend for S5 seems to refer to the matching Figure 6.)

15. page 25. Since gene expression control is mediated by RBPs.... This is too narrow. By now there is ample evidence that epigenetic and transcription factors are also important.

Modified

16. It is not clear why standard terms are often put in inverted commas.

Two are in a legend and I think should stay; the third set has been removed.

17. There are a few typos in the manuscript, e.g. urudyliyl, because...

These two have been fixed and I have checked the rest of the paper as well.

Figures

Figure 1. It is not clear why "metacyclic" is written in grey, and all other stages in black.

It's in the legend: Non-dividing transmission forms are labeled in grey

Figure 2. The author is entitled to reproduce a figure that she has published before – no problems with that. The issues here are that it is extremely complicated and does not reflect recent findings (e.g. co-

transcriptional nuclear export of mRNA, Goos et al., 2018, participation of RHS in transcription). On the same page as the figure (p.3) processing is referred to as post-transcriptional.

"Post-transcriptional" tends to be used by RNA people to distinguish between "co-transcriptional" processes that are obligatorily coupled to transcription, such as capping, and processes that happen after at least most of the mRNA has been made. But I agree that in the context of splicing, it is confusing. The Goos et al. paper certainly shows that co-transcriptional export can happen but I suspect that it is unusual. The mRNA for which it was shown is one of the longest in the cell, 22 kb, and the transcripts were stuck because *trans* splicing had been inhibited.

Wouldn't it be more appropriate to have a figure summarising what is known about mRNA transcription by RNAPI and II?

The review barely covers pol I and would be impossibly long if it was added - see notes above.

Figure 3 is also very complicated for a general audience.

I've made it as simple as possible.

The majority of splicing and processing events are of type B, not type C. One could leave it at that.

I don't think we know that.

Figure 4. I could not work out which results were from *Leishmania* and which were from *T. brucei*. Maybe this could be colour-coded.

This has been clarified in the legend.

At the same time, the activator/repressor colour code could be removed. These data come from tethering assays and it is not clear if they also apply under normal circumstances.

So far these have held up pretty well. We have only, so far, found one instance in which the tethering result may not reflect the real activity of the protein. It's known that

Unpublished data from the author's lab is apparently included. It is not clear how extensive this is in this particular figure - it will also depend on the journal's policy (see point 6).

This has been clarified in the legend. I have also removed two potential pull-down partners of 4EIP from the Figure since there were several others and the specificity of all of them was dubious.

Figure 5. It would be helpful to use the same colours in this figure and figure 1.

Good idea. Done.

It is not clear where the expression data come from. Do they refer to RNA or protein?

How much is published/unpublished?

Sorry, I meant to add the references to the legend, but forgot. It has now been done.

The different scales of blue and grey are not so obvious on a printout – increasing contrast or employing different colours would help.

The table has been considerably simplified.